# LLM4DV: Using Large Language Models for Hardware Test Stimuli Generation

## Abstract

Hardware design verification (DV) is a process that checks the functional equivalence of a hardware design against its specifications, improving hardware reliability and robustness. A key task in the DV process is the test stimuli generation, which creates a set of conditions or inputs for testing. These test conditions are often complex and specific to the given hardware design, requiring substantial human engineering effort to optimize. This leads to a significant challenge in automated and efficient testing for arbitrary hardware designs. We seek a solution that takes advantage of large language models (LLMs). LLMs have already shown promising results for improving hardware design automation, but remain under-explored for hardware DV. In this paper, we propose an open-source benchmarking framework named LLM4DV that efficiently orchestrates LLMs for automated hardware test stimuli generation. Our analysis evaluates six different LLMs involving six prompting improvements over eight hardware designs and provides insight for future work on LLMs development for efficient automated DV.

## 1 Introduction

Large Language Models (LLMs) (Yang et al., 2020; int, 2020; Touvron et al., 2023) have gained significant attention in recent years due to their language generation and comprehension capabilities on tasks such as language translation (Feng et al., 2020), question answering (Yang et al., 2020), and sentiment analysis (Liu et al., 2021). Recently, there has been interest in exploiting LLMs to improve hardware design generation (Blocklove et al., 2023; Fu et al., 2023; Lu et al., 2024). Arguably, hardware design verification (DV), which checks the correctness of hardware designs, ranks among the most crucial and time-consuming tasks in hardware development. Hardware DV is often ***time-consuming***, usually taking up to 60%-70% of the development time (Shin, 2024), and requires significant ***human guidance and expertise*** due to the complexity of both hardware design and its corresponding testing requirements (Shin, 2022).

On the other hand, existing work on LLMs has been studied for software testing. For example, Codex (Chen et al., 2021) can produce functionally correct bodies of code from natural language docstring descriptions. LLaMA 2 (Touvron et al., 2023), an LLM using instruction tuning and Reinforcement Learning with Human Feedback (RLHF) (Christiano et al., 2017; Stiennon et al., 2020) for fine-tuning, emerges impressive generalization and external tool usage ability. However, these approaches are not directly applicable due to the following two challenges. First, unlike software programming languages, there is a scarcity of high-quality, open-source hardware designs and testing code available online for training LLMs. This limitation is critical because Hardware Description Languages (HDLs) possess ***distinct semantics*** that differ fundamentally from software programming languages. These unique characteristics make HDLs considerably more challenging for LLMs to interpret and learn from, as the models cannot simply transfer their knowledge from conventional programming contexts without substantial modifications. Second, the testing space for a hardware design design is typically large, leading to a ***scalability*** problem. Existing approaches on hardware DV require human guidance to reduce search space, such as adding heuristics to guide tests of a particular hardware design. This raises an important question: ***can LLMs effectively minimize the amount of human effort involved in hardware DV?***

In this work, we specifically focus on hardware test stimuli generation, which generates test inputs for hardware DV. In the DV process shown in the right of Figure 1, the test stimuli generation stands out as the most labor-intensive phase, often requiring iterative trial-and-error. A good stimuli discovers

Figure 1: An overview of LLM4DV framework. The *right part* shows a traditional DV process. DV engineers need to manually interact with the DV process by tailoring various stimulus and observing the coverage. Such a manual process is often iterative. The *left part* highlights our contributions, which adds the stimulus generation agent for automated guidance.

new hardware states during testing, increasing the test coverage; while a bad stimuli only tests existing states, leaving the coverage the same. Finding good stimuli becomes particularly arduous when encountering hard-to-hit points within the coverage plan. In order to find a path to LLM solutions, we present a novel benchmarking framework named LLM4DV (Large Language Model for Design Verification), that utilizes LLMs for **test stimuli generation**, and make the following contributions:

- We introduce and construct LLM4DV, a framework that employs prompted LLMs to generate test stimuli for hardware DV. Our complete workflow facilitates a plug-and-play flow for users to experiment various LLMs, hardware designs and test coverage plans. We show automated DV requires a complex prompting strategy and also propose six prompt enhancements to establish strong baselines for the LLM4DV framework. We believe this provides an attractive testbed for experimenting the agentic behavior of LLMs.

- We design and construct three DUT modules: a Primitive Data Prefetcher Core, an Ibex Instruction Decoder, and an Ibex CPU. We also select five open-source designs, obtaining a varied set of DUTs with different testing difficulties that are supplied within the LLM4DV flow for users.

- We evaluate LLM4DV using these eight DUT modules and introduce a set of evaluation metrics. We show that LLMs, with optimized prompt enhancements, achieve coverage rates (a primary metric for measuring verification effectiveness) ranging from 89.74% to 100% in a realistic setting. We open-source LLM4DV alongside these modules to allow both the machine learning and hardware design communities to experiment with their ideas.

The rest of the paper is organized as follows. Section 2 provides a background of traditional hardware DV processes. Section 3 reviews related work in the field of LLM-assisted software testing and digital hardware design. Section 4 describes LLM4DV in detail. Section 5 evaluates the effectiveness of several LLMs inside the framework.

## 2 BACKGROUND

A traditional hardware DV process is illustrated in the right of Figure 1. For each hardware design, also known as device-under-test (DUT), the hardware designer provides a functionally equivalent golden model in software to the DUT (Witharana et al., 2022). The DV process takes a set of inputs, or test stimuli, and sends them to both the DUT and its golden model (①), leading to two sets of results. The results are then compared between the DUT and its golden model (②). If the results are identical, the DUT behaves correctly in the hardware states triggered by the **current test stimulus**, leading to a coverage of verified states. In order to progressively expand the testing coverage, a DV process typically tests the DUT iteratively on a large set of stimuli defined by the hardware designer in advance. These stimuli aim to cover a wide range of scenarios and use cases that the hardware might encounter in real-world applications, which are defined in the **coverage plan** in the form of coverage bins. A coverage bin is a specific condition or scenario that the verification environment tracks to determine whether a particular aspect of the design has been exercised or tested. A number of bins are defined in a coverage plan for each value of interest. For simplicity, all coverage points

Table 1: Comparison to related work applying LLMs in the field of digital hardware.

| Name | Task | Number of models | Testing space |
|---|---|---|---|
| RTLFixer(Tsai et al., 2023) | Verilog syntax correction | 1 | 212 syntax errors |
| NSPG(Meng et al., 2023) | Repairing security-relevant bugs in Verilog | 4 | 10 designs (10 bugs) |
| ChipNeMo(Liu et al., 2023) | Bug analysis and summarisation | 3 | 30 bugs |
| Kande et al.(Kande et al., 2023) | Generating security assertions | 4 | 10 designs (10 assertions) |
| Thakur et al.(Thakur et al., 2023) | Generating Verilog code | 6 | 17 problems |
| RTLLM (Lu et al., 2023) | Generating Verilog code | 4 | 30 designs |
| LLM4DV | Stimulus generation for functional verification | 5 | 8 designs (3883 coverage bins) |

Table 2: Comparison to related work with non-LLM hardware testing techniques

| Features | (Fine and Ziv, 2003a) | (Braun et al., 2004) | (Baras et al., 2009) | (Fine et al., 2005) |
|---|---|---|---|---|
| Models | Bayesian | Bayesian | Bayesian | Bayesian |
| Encoding model | MAP | MAP | MAP | MAP |
| Retraining | Yes | Yes | Yes | Yes |
| **Features** | **(Gal et al., 2021)** | **(Gal et al., 2020)** | **(Vasudevan et al., 2021)** | **Ours** |
| Models | DNN | DNN | GNN | LLM |
| Encoding model | gradient-based | gradient-based | gradient-based | text |
| Retraining | Yes | Yes | Yes | No |

are considered to only include a single coverage bin in this work. The coverage monitor (③) inspects the DUT's inputs, outputs, and internal states; determines whether there are hits of coverage bins; updates the current coverage and returns it to the stimulus generation agent for the next stimulus. The procedure in the right of Figure 1 typically follows an iterative approach, often executed tens of thousands of times, in which a human DV engineer applies various stimuli to achieve comprehensive coverage specified in the coverage plan.

Effective test stimuli generation has been a major challenge in meeting 100% coverage (Witharana et al., 2022). For a simple design, verification can be done with individual directed tests, in which test stimuli (inputs for the DUT) are manually generated. For more complex designs, a large number of stimuli is required for exercising as much of the design's functionality as possible. Traditionally, **constrained-random testing (CRT)** has been used to generate vast random but valid test stimuli and to attempt to "hit" the bins. However, CRT is inefficient to hit as many bins as human effort for hardware states with complicated conditions. Still, it remains the case that extensive human engineering involvement is required for the test stimuli design process.

## 3 RELATED WORK

While the application of LLMs on hardware design verification has been brought to focus only recently, test code generation for software engineering with LLMs has been well-studied and achieved remarkable performance (Chen et al., 2022; Schäfer et al., 2023; Lukasczyk and Fraser, 2022). Chen et al. (Chen et al., 2022) utilize LLM to suggest potential implementations and corresponding test cases for a function. They provide the LLM with the signature and a description of the function and select the best solution based on functionality agreement. Schäfer et al. (Schäfer et al., 2023) propose a pipeline to generate unit tests for existing code, which iteratively refines the prompt to generate better tests. These studies show promising results on software code, while we shift focus to using LLM to reason HDLs and hardware design specifications, leading to a different setting and more sophisticated procedure.

In hardware design verification, assertion-based verification (ABV) is also widely adopted together with code coverage, functional coverage, and validation using generated test patterns (Witharana et al., 2022). ABV inserts assertions into the DUT HDL source to detect violations of predefined design properties. However, ABV requires test patterns (i.e. input test stimuli) to activate given assertions and therefore reveal vulnerabilities. For simulation-based ABV approaches, traditional test generation that uses random or constrained-random tests cannot guarantee to activate assertions with complex conditions in a reasonable time. In order to speedup ABV, Pal et al. (Pal et al., 2008) propose bias random test generation. They consider the DUT as a black box and restrict test generation to only input/output signals. Ferro et al. (Ferro et al., 2008) used combinatorial testing, which provides

Table 3: A list of hardware DUTs and their golden models provided by the LLM4DV benchmark set.

| Names | Descriptions |
|---|---|
| Primitive Data Prefetcher Core* | Detects stride patterns in a series of integers. Limited logical reasoning abilities are required to achieve high coverage. |
| Ibex CPU Instruction Decoder* | Decodes RISC-V Instruction codes. Detailed knowledge about the RISC-V ISA is needed to achieve full coverage. |
| Ibex CPU* | A RISC-V CPU core. Detailed knowledge about the RISC-V ISA, as well as familiarity with CPU architecture is needed to achieve full coverage. A brief description of relevant instructions, as well as the format of R-, S-, and J-type instructions is given to the agents in the initial prompt to enhance performance. |
| Asynchronous FIFO (Pretet) | A dual clock FIFO used to exchange data between clock domains. Coverage bins are straightforward, but the LLM has to control inputs to two clock domains simultaneously. |
| AMPLE Prefetcher Weight Bank (Gimenes) | A component of AMPLE, a graph neural network accelerator. It is responsible for fetching data from memory, storing it in a FIFO, and outputting it in a diagonal form. There is a direct link between each possible input combination and a corresponding coverage bin, no significant reasoning abilities are needed to achieve high coverage. |
| AMPLE Prefetcher Fetch Tag (Gimenes) | Another component of AMPLE. Similar to the weight bank, its basic purpose is to fetch data from memory. However, it contains not entirely independent queues that need to be managed simultaneously. The device is simple, but there is no clear link between input combinations and coverage bins, requiring significant reasoning abilities. |
| SDRAM Controller (Horne) | A very simple SDRAM controller. The LLM may need to be pre-trained with knowledge about SDRAM control signals, or needs to infer them from the device's source code (if given) to achieve full coverage. |
| MIPS CPU (TrivialMIPS) | A MIPS CPU core. Detailed knowledge about the MIPS ISA, as well as familiarity with CPU architecture is needed to achieve full coverage. Similar to the Ibex CPU, a brief description of relevant instructions, as well as the format of R-, I-, and J-type instructions is given to the agents in the initial prompt to enhance performance. |

a set of combinations of user-selected values, to select test stimuli most suitable to cover corner cases. Tong et al. (Tong et al., 2009) propose a method that searches for compact assertion-based automata for failure and acceptance nodes before test generation. Simulation-based test generation has also been incorporated with formal method-based test generation. Lyu and Mishra (Lyu and Mishra, 2020) utilized concolic testing to activate assertions. They consider assertions as branches, search through the branch statement tree with heuristics to efficiently obtain a path, and generate test stimuli to cover the obtained branch targets. These methods, however, are subjected to complexity explosion problems (Witharana et al., 2022) and fail to effectively make use of the user's knowledge about the design. We overcome this issue by utilizing LLM's pre-trained knowledge to reason about the given coverage plan and guide the test stimuli generation. Other advanced testing techniques, such as coverage-directed generation and mutating tests (Fine and Ziv, 2003b; Guzey and Wang, 2007; Laeufer et al., 2018), have been studied to improve the performance of CRT. These works may guide test generation by LLMs to achieve better results but face challenges in the integration out of the scope of this paper. Our LLM4DV framework provides an initial platform and baselines for their evaluation in future works. **@Reviewer 1C9a:** Further works that approach test generation with non-LLM machine learning methods are summarized in Table 2.

Recently, the application of LLMs for hardware design and verification purposes has started to gain traction (Zhong et al., 2023). Table 1 provides a summary of recent benchmarks that focus on applying LLMs within this domain. In particular, *there are currently no benchmarks that evaluate the stimuli generation capabilities of LLMs.* Among the recent contributions, RTLFixer (Tsai et al., 2023) enables the automated correction of Verilog syntax errors. In contrast, NSPG (Meng et al., 2023) is designed to extract security properties by analyzing hardware documentation. ChipNeMo (Liu et al., 2023) has been assessed on tasks related to bug summarization and analysis. Additionally, Kande et al. (Kande et al., 2023) proposed a methodology for automatically generating SystemVerilog assertions (SVAs) using LLMs to enhance hardware security. Meanwhile, Thakur et al. (Thakur et al., 2023) and RTLLM (Lu et al., 2023) have explored the generation of Register Transfer Level (RTL) code using LLMs. While it is challenging to directly compare the scale of these benchmarks with that of LLM4DV due to the different abilities assessed, it should be noted that LLM4DV's scope of 3883 coverage bins across 8 devices, tested with six different off-the-shelf models, represents a significant contribution to the field.

## 4 LLM4DV Benchmarks

Our experiments use an LLM in the test stimuli generation process, together with a testbench containing a DUT to form the complete LLM4DV framework. The following subsections describe the basic DV framework, the prompt templates for the LLM, and six prompting improvements. Figure 1 gives a general picture of the prompt templates and prompting improvements.

Table 4: A list of input options and output evaluation metrics for the proposed LLM4DV framework.

| | Names | Descriptions |
|---|---|---|
| Input Options | DUT | The target DUT to be tested. |
| | Model | The LLM used for stimulus generation. |
| | Prompting Configurations | The prompting strategy used for stimulus generation. |
| | Coverage Plan | The coverage plan specified for the target DUT. |
| Evaluation Metrics | Max Coverage | Maximum recorded number of coverage bins (defined by the coverage plan) covered. A higher number indicates better performance. |
| | Effective Message Count | The minimum number of messages an LLM produces across several trials in an experiment achieving maximum coverage; a lower count indicates better performance. |
| | Average Message Count | Average number of query messages per experiment $\pm$ standard deviation of messages. As the usage of LLMs is costly, a faster convergence to maximum coverage is preferred. |

## 4.1 LLM4DV Framework

In this work, the proposed LLM4DV framework automates the DV process by exploiting LLMs for test stimuli generation, shown in the left of Figure 1. Compared to traditional DV processes that use user-defined test stimuli, the stimulus generation agent uses an LLM to provide a test stimulus in each timestep. This reduces human involvement in the hardware DV loop and effectively guides tests to increase coverage rates.

In each generation cycle, the prompt generator produces a prompt based on a template (⑦) and the current coverage feedback from the coverage monitor. LLM4DV allows customization of prompts inside a dialogue, this means each *query message* can receive different prompts, as managed by the query scheduler (⑤) shown in Figure 1. This is explained in Section 4.3.

The LLM takes in the prompt and generates a natural language response, from which the test stimulus values are extracted and sent to the DV flow in the right of Figure 1. The DV framework then produces current coverage which is considered as input for the LLM-based stimulus generation agent (⑧) shown in Figure 1. The processes of test stimuli generation and hardware testbench simulation are executed in parallel asynchronously. Specifically, a buffer is placed between the stimulus generation agent interfaces to balance the rate of the test stimuli generation and consumption. In every timestep when the stimulus generation agent is requested for a test stimulus, it takes out the oldest value in its stimulus buffer; if the buffer is empty, the agent starts a new generation cycle, in which the LLM takes in a new request and a list of new stimuli will be added to the buffer.

In LLM4DV, each DV process is viewed as a "***trial***", where there would be multiple dialogues made in a single trial, as illustrated in Figure 1, which are controlled by the dialogue scheduler (⑥). A *trial* stops in one of the following three states, and the agent is considered "exhausted". When reaching such a state, it becomes ineffective or inefficient to expand testing coverage and the pipeline stops. First, full coverage is reached, where all coverage bins have been hit. Second, no new coverage is extended within a number of trials, where our implementation by default specifies that the stimulus generation agent cannot hit any bins within 25 responses. Finally, the coverage expansion speed is low, where our implementation by default specifies that the stimulus generation agent hits fewer than three bins within 40 responses. **@Reviewer 9kfb:** Algorithm 1 provides the exact implementation of the pipeline. The exact states can be specified by users as input to the framework, and here we use the particular setups above for fair evaluation across DUTs, LLMs and prompting methods.[1] Within the LLM4DV framework, we explore six prompting strategies and improvements over a set of LLMs and DUTs. We describe our evaluation method in Section 4.

## 4.2 Evaluation Setup

The proposed LLM4DV benchmark contains eight DUT modules, as listed in Table 3. Three of the devices were developed by the authors, and the other five are open-source designs. These DUTs are selected because they are commonly seen in most representative computer architectures such as CPUs, GPUs and other hardware accelerators. Detailed information about the DUTs is provided in Appendix A.4. We use six different commercially available LLMs: GPT-3.5 Turbo, Llama v2 70B Chat, Claude 3 Sonnet, CodeLlama 70B Instruct, Llama 3 70B Instruct, and Claude 3.5 Sonnet. To evaluate the effectiveness of these LLMs, we observe the testing performance based on three

---

[1]See ~~Algorithm 1 for exact implementation, and~~ Appendix A.2 for a justification of these hyperpramters.

---

**Algorithm 1** Basic DV Pipeline

---

stimulus ← 0
coverage ← {}
**while** coverage rate < 100% and not (Δ coverage in 25 messages = 0 or Δ coverage in 40 messages < 3 **do**
    **while** stimulus_buffer not empty and coverage rate < 100% **do**
        stimulus ← stimulus_buffer.pop()
        testbench.input(stimulus)
        coverage←coverage_monitor.compute_coverage(testbench)
    **end while**
    prompt ← prompt_generator.generate(coverage)
    response ← LLM.generate(prompt)
    stimuli ← extractor.extract(response)
    stimulus_buffer.extend(stimuli)
    **while** stimulus_buffer is empty **do**
        prompt ← prompt_generator.regenerate(coverage)
        response ← LLM.generate(prompt)
        stimuli ← extractor.extract(response)
        stimulus_buffer.extend(stimuli)
    **end while**
**end while**

---

evaluation metrics, as listed in the lower part of Table 4. We have limited each trial to 700 messages. The design choices of these parameters are explained in Appendix A.1.

## 4.3 GENERAL PROMPTING STRATEGIES

We provide a Coverage-Feedback Template to generate prompts for the LLM. When constructing it, we utilize prompt engineering techniques including 1) System message: it is included at the beginning of every prompt, and is used to prime the model with context, instructions, or other information relevant to the use case; 2) Start with clear instructions; 3) Repeat instructions at the end; 4) Add clear syntax: punctuation, headings, and section markers; 5) Specifying the output structure.

**Coverage-Feedback Prompt Template** The Coverage-feedback prompt template contains templates for the system message, initial query, and iterative queries.

- The **system message** clarifies the expected response format and specify other requirements.

- The **initial query** is the first user query message in a dialogue. It contains three parts: 1) Task introduction: a description of what is included in this prompt and what the LLM will be asked to do; 2) Coverage plan summary: a description of cover bins of the coverage plan; and 3) Initial question: a one-line instruction.

- The **iterative queries** are the user messages following the first assistant (LLM) response. Each contains three parts:

  1. Result summary: a general feedback which:
     - if the previous assistant response was gibberish (i.e. contains mostly nonsense words) or didn't follow the output format, the result summary repeats the output format requirement;
     - otherwise if the previous assistant response failed to hit any new bins, the result summary points that out and ask for a new list of stimuli;
     - if the previous assistant responses hit some bins, the result summary points that out and ask for a new list of stimuli.
  2. Differences: a list of uncovered bins.
  3. Iterative question: a one-line instruction, repeating the output format requirement if previous response was gibberish or didn't follow the output format.

## 4.4 FOUR GENERIC PROMPTING IMPROVEMENTS

In our experiments, we develop two improvements necessary for making the framework executable and two improvements that increase its performance on most cases, which can be effectively employed regardless of the nature of the DUT and coverage plan. Details see Appendix A.3.

**Missed-bin Sampling** This optimization is ④ in Figure 1, and is later used in the query scheduler. In most generation cycles in a trial, there would be hundreds to thousands bins uncovered. The iterative queries can't include all of them because the prompt's length would exceed the LLM's input token number limit. Meanwhile, exposing too many uncovered bins to the LLM confuses the LLM on which mistakes should it resolve first.

We propose missed-bin sampling, which samples a number of bins from all uncovered bins to be included in the differences part of iterative queries. Our experiment finds that more random sampling methods encourage the agent to cover bins with stricter hitting conditions, and more stable sampling methods make the agent more efficient in hitting the easier bins.

We define three sampling methods (1) Pure Random Sampling, which randomly samples seven bins from all uncovered bins. (2) Coverpoint Type-based Sampling, which samples from "easier bins" and "harder bins" respectively. (3) Mixed Coverpoint Type-based and Pure Random Sampling, which switches between previous two whenever the agent becomes inefficient with current strategy.

**Best-iterative-message Sampling** The LLM needs previous query messages in the dialogue to learn about what has happened. However, as the dialogue grows, the length of input may exceed the LLM's input limit. One solution is summarizing previous query message, which helps generalizing concepts in the dialogue but loses details, which is crucial in our task. On the other hand, sampling from previous messages acceptably loses some generality meanwhile preserves key details, including the bin description and positive examples (i.e responses that successfully hit many bins) useful for covering corner cases. These strategies are used in our Query Scheduler in ⑤.

We propose four sampling methods (I) Recent Responses, where we keep the initial query (and its response), and three most recent iterative queries (and their responses). (II) Successful Responses, where we randomly keep three that hit the largest number of bins. (III) Mixed Recent and Successful Responses, where we keep two most successful and one most recent query. (IV) Successful Difficult Responses, which is similar to Successful Responses but each "harder bin" counts as 2.5 bins.

**Dialogue Restarting** LLMs sometimes behave stubbornly, repeating mistakes they made previously. We introduce a dialogue restarting scheduler (⑥) to resolve this problem. When the LLM hits less than three new bins within t responses, we clear the dialogue record and restart from the system message and initial query. We define four dialogue restarting schedules (a) Normal Tolerance, where t = 7. (b) Low Tolerance, where t = 4. (c) High Tolerance, where t = 10. (d) Coverage Rate-based Tolerance, where t = 4 in the beginning and t = 7 after reaching certain coverage rate threshold.

**Best-iterative-message Buffer Reset** When the dialogue record is reset, the buffer for best iterative messages in Best-iterative-message sampling can also be cleared or kept. These two strategies display a trade-off between "effectively forgetting past mistakes" and "learning about the task faster after restart". This reset is also then incorporated in the dialogue restarting scheduler (⑥).

We define three resetting plans for the best-iterative-message buffer, (i) Clearing best-messages. (ii) Keeping best-messages. (iii) Stable-restart Keeping best-messages: keeping the buffer on restarts, but not using it for the first four responses after restarts.

We have employed distinct notations to denote the available options for these prompting enhancements. This enables us to encode specific combinations, such as Claude-3 1 I a i, indicating that the first option is selected for all the aforementioned prompting strategies.

### 4.5 TWO SITUATIONAL PROMPTING IMPROVEMENTS

We have developed two additional prompting improvements, which can be effectively deployed depending on the nature of the DUT and the coverage plan.

**Providing the DUT Code** By including the DUT source code in the initial message and incoporate this change in the query scheduler (⑤), we try to enhance the model's performance with context-specific information that is intrinsic to the device's operational logic and architecture. By parsing the HDL code, the LLM may directly correlate specific features and functions with the corresponding coverage bins, ensuring that the generated stimuli are not only syntactically correct but also semantically aligned with the DUT's functional requirements. However, due to bounded context windows, this technique may only be employed for devices with limited source code length.

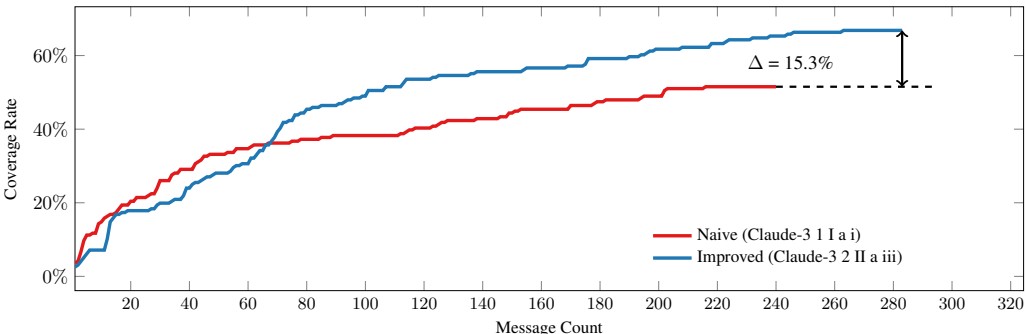

Figure 2: Performance improvement due to the employment of the four generic prompting strategies on the IBEX CPU, using Claude 3 Sonnet.

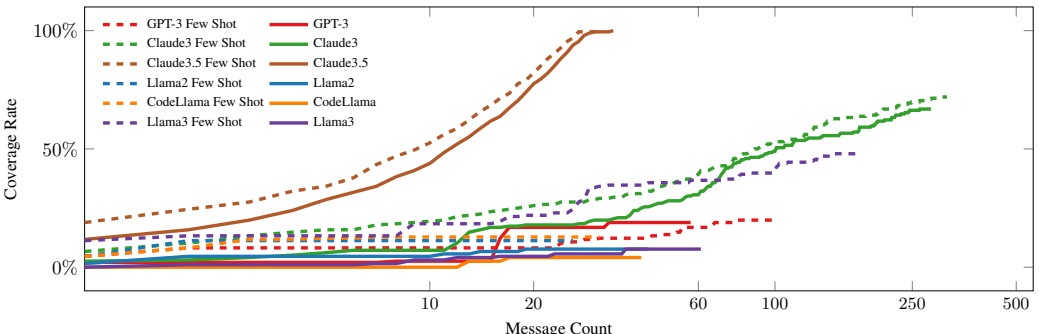

Figure 3: Performance improvement due to few-shot prompting on the IBEX CPU.

**Few-shot Prompting** As task-specific fine-tuning is outside the scope of this study, we instead aim to use few-shot prompting to improve coverage metrics. By including in the initial prompt a few examples of stimuli generating bin hits in the query scheduler (⑤), the LLMs may adapt to the context of hardware verification, and assimilate some information about the DUT. To avoid skewing the experimental results, this has only been employed in cases where the coverage plan includes a significant amount of coverage bins.

## 5 RESULTS AND ANALYSIS

**Using the Generic Prompting Strategies** We ran our experiments on Intel Xeon CPUs using LLM APIs hosted on a platform named OpenRouter. The total cost for OpenRouter was USD334. For each reported result three experiments were performed. In Section 4.4, we introduced these four generic prompting techniques, each accompanied by several configurations: missed-bin sampling (choices (1)-(3)), best-iterative-message sampling (choices (I)-(IV)), dialogue restarting (choices (a)-(d)), and best-iterative-message buffer reset (choices (i)-(iii)). Figure 2 shows coverage rate gains when correctly employing the four strategies. The naive approach is considered to be the simplest configuration: (1) Random Sampling, (I) Recent Responses, (a) Normal Tolerance, and (i) Clearing best-messages. Through extensive experimentation across different configurations detailed in Appendix A.6, the best configuration was identified as (2) Coverpoint Type-based Sampling, (II) Successful Responses, (a) Normal Tolerance, and (iii) Stable-restart Keeping best-messages. While the naive configuration only achieved 51.53% coverage, the chosen strategy reached 66.84%, an increase of 15.31%. In all further experiments, this generic prompting configuration is used.

**Providing the DUT Code** The varying effects of providing the DUT's HDL source code to different LLMs are shown in Table 6 in our Appendix. Out of the 20 LLM-DUT pairs, only in 9 cases can we observe an increase in performance when providing the DUT source code. In all other cases, degradation occurs in terms of both maximum achieved coverage rates and trial lengths. This is likely due to the limited context size of the LLM agents. Whether this prompting strategy leads to benefits

Table 5: Best results achieved for each LLM-DUT pair. In all cases, the generic prompting strategy described in Section 5 was used. Additionally, experiments marked with * used few-shot prompting, and experiments marked with † included the DUT source code in the initial prompt. We highlight the **best** results for each DUT. Note that trials were limited to 700 messages.

| | | Primitive Data Prefetcher Core | Asynchronous FIFO | AMPLE Prefetcher Weight Bank | AMPLE Prefetcher Fetch Tag |
|---|---|---|---|---|---|
| gpt-3-turbo | Max coverage | 1016 (98.26%)* | 10 (100%) | 324 (100%)† | **10 (100%)** |
| | Eff. msg. count | 350 | 16 | 36 | **2** |
| | Avg. msg. count | 509.0 ± 129.4 | 19.7±3.9 | 37.7±1.2 | **22.0±14.1** |
| llama-2-70b-chat | Max coverage | 431 (41.68%)* | 10 (100%)† | 324 (100%) | 10 (100%)† |
| | Eff. msg. count | 700 | 1 | 36 | 22 |
| | Avg. msg. count | 470.7±189.9 | 10.5±7.9 | 41.3±7.5 | 27.7±6.0 |
| claude-3-sonnet | Max coverage | 801 (77.47%)* | **10 (100%)** | 324 (100%) | 10 (100%) |
| | Eff. msg. count | 700 | **1** | 36 | 8 |
| | Avg. msg. count | 676.3±33.5 | **1.0** | 36.0 | 19.3±8.0 |
| codellama-70b-instruct | Max coverage | 82 (7.93%)* | 10 (100%) | 324 (100%) | 6 (60.00%)† |
| | Eff. msg. count | 154 | 1 | 44 | 34 |
| | Avg. msg. count | 102.0±50.3 | 3.7±3.1 | 52.3±8.5 | 28.3±4.0 |
| llama-3-70b-instruct | Max coverage | 710 (68.67%)* | 10 (100%)† | **324 (100%)** | 10 (100%)† |
| | Eff. msg. count | 700 | 1 | **26** | 15 |
| | Avg. msg. count | 700.0 | 1.3±0.5 | **32.7±4.7** | 20.0±3.6 |
| claude-3.5-sonnet | Max coverage | **1022 (98.84%)*** | **10 (100%)** | 324 (100%) | 9 (90%) |
| | Eff. msg. count | **321** | **1** | 36 | 25 |
| | Avg. msg. count | **329.3±32.3** | **1.0** | 36.7±0.6 | 25.0 |
| Formal verification | Max coverage | 1030 (99.61%) | 10 (100%) | 3 (0.93%) | 10 (100%) |
| CRT | Max coverage | 0 (0%) | 10 (100%) | 324 (100%) | 10 (100%) |

| | | SDRAM Controller | Ibex CPU Instruction Decoder | Ibex CPU | MIPS CPU |
|---|---|---|---|---|---|
| gpt-3-turbo | Max coverage | 7 (100%) | 1466 (69.58%)* | 39 (19.90%)* | 84 (43.08%)* |
| | Eff. msg. count | 7 | 700 | 102 | 211 |
| | Avg. msg. count | 22.3±11.0 | 432.0±228.3 | 88.0±21.2 | 111.0±72.8 |
| llama-2-70b-chat | Max coverage | 6 (85.71%)† | 402 (19.08%)* | 22 (11.22%)* | 68 (34.87%)* |
| | Eff. msg. count | 32 | 186 | 26 | 55 |
| | Avg. msg. count | 28.3±2.6 | 125.7±61.1 | 33.3±10.4 | 45.7±13.2 |
| claude-3-sonnet | Max coverage | 7 (100%)† | 1512 (71.76%)* | 141 (71.94%)* | 159 (81.54%)* |
| | Eff. msg. count | 2 | 700 | 315 | 299 |
| | Avg. msg. count | 2.3±0.5 | 700.0 | 287±19.9 | 277.7±35.2 |
| codellama-70b-instruct | Max coverage | 7 (100%)† | 417 (19.79%)* | 25 (12.76%)* | 91 (46.67%)* |
| | Eff. msg. count | 8 | 182 | 31 | 142 |
| | Avg. msg. count | 29.3±15.1 | 126.3±57.6 | 34.3±6.9 | 113.7±20.4 |
| llama-3-70b-instruct | Max coverage | **7 (100%)** | 1135 (53.89%)* | 94 (47.96%)* | 98 (50.26%)* |
| | Eff. msg. count | **1** | 700 | 172 | 175 |
| | Avg. msg. count | **2.3±1.2** | 700 | 180.3±20.9 | 141±24.1 |
| claude-3.5-sonnet | Max coverage | 7 (100%)† | **2006 (95.21%)*** | **196 (100%)*** | **175 (89.74%)*** |
| | Eff. msg. count | 2 | **651** | **31** | **176** |
| | Avg. msg. count | 2.0 | **683.7±28.3** | **37.0±5.29** | **174.7±41.0** |
| Formal verification | Max coverage | 7 (100%) | 2106 (99.95%) | 100% | 100% |
| CRT | Max coverage | 7 (100%) | 1154 (54.77%) | 30 (15.31%) | 28 (14.36%) |

depends on the specific LLM agent and DUT, so the decision to employ it needs to be decided on a case-by-case basis.

**Few-shot Prompting** The LLMs were given specific examples of stimulus-coverage bin hit pairs in experiments where the coverage plan includes more than 20 bins. The specific number of examples was chosen empirically depending on the variety of coverage bins, but in all cases between 5 and 10. Figure 3 compares the performance of all six LLMs when tested on the Ibex CPU, where dashed lines represent trials with few-shot prompting enabled. The models reach completion at varying message

counts due to the stop condition outlined in Section 4.1. The prompting guides the model to verify the design more efficiently. In all the cases, significant improvement is observed in terms of coverage rates observed, when few-shot prompting is applied. Among these LLMs, Claude 3.5 shows the best results, where both zero-shot and few-shot approaches reached full coverage.

**Final Results** Table 5 presents the best results achieved for each LLM-DUT pair, compared with naive CRT and formal methods serving. In the CRT methodology, we generate 100,000 combinations within the valid input range without additional constraints. The formal baseline utilizes the cover mode of the SymbiYosys tool (SymbiYosys), where all bins of the coverage plans correspond to specific SystemVerilog cover statements, and each formal verification run is limited to a 48-hour timeout.

Across all DUTs, each configuration demonstrates that LLM4DV can either match or exceed the coverage rates achieved via naive CRT. This signifies not only the adaptability of LLMs to varied hardware testing contexts but also their potential to streamline certain aspects of verification by reducing reliance on extensive random input generation. Formal methods only work well when the design states are small. Particularly, the AMPLE Prefetcher Weight Bank only achieves a coverage lower than 1%, because it contains large storage queues, which introduces a large number of feasible states to represent values in all possible orders. In fact, the number of states grows exponentially with queue size, leading to a "state space explosion", despite the simplicity of the design.

In Table 4, our evaluation metrics encompass not just the maximum coverage rate but also the maximum and average message counts. This comprehensive evaluation becomes particularly valuable when maximum coverage attained is $100\%$, which could happen with less complex DUTs. Maximum and average message counts allow for assessing the efficiency of LLMs in achieving this state of success. Practically speaking, this aspect is beneficial as a more expedient DV cycle is normally desirable.

Among the LLM models tested, Claude 3.5 Sonnet stands out, especially in handling more complex tasks such as those associated with CPU architectures. This model's superior performance in scenarios involving the Ibex and MIPS CPU may indicate a more nuanced understanding of CPU operations, likely stemming from richer pre-training that possibly included diverse computational and hardware-related datasets. Claude 3.5 Sonnet's effectiveness in these settings could suggest that its training included exposure to architectural nuances specific to CPUs, enhancing its ability to generate more relevant and coverage-effective test stimuli. Nevertheless, it still falls short of the $100\%$ mark achieved by the formal tool. This suggests that while LLMs can handle complex scenarios to a degree, they may lack the deep, specialized knowledge or the ability to effectively navigate the vast state spaces that high-complexity DUTs entail.

The consistently high coverage achieved by all LLM models in testing lower complexity DUTs, such as the Asynchronous FIFO and SDRAM Controller, demonstrates the proficiency of LLMs in handling straightforward scenarios. This high performance is mirrored by the baseline formal tool, indicating that LLMs are competent and can rival traditional verification methods in simpler verification contexts. Claude 3 Sonnet, for instance, maintains $100\%$ coverage across simpler DUTs, suggesting excellent efficiency in generating relevant test cases with minimal extraneous inputs. The efficiency of test generation, as reflected by the message count metrics, provides another dimension of evaluation. Models like Claude 3 Sonnet, which generally require fewer messages to achieve high coverage, indicate a more targeted and efficient approach to test case generation. In contrast, models requiring a higher number of messages, such as Llama-2-70b-chat and Codellama-70b-Instruct, may be generating less precise or less effective test stimuli, indicating inefficiencies that could translate to increased testing time and resource consumption in practical applications.

## 6 DISCUSSION

**Gimmick or Trend?** The computer architecture and hardware design community is now starting to see debates regarding the effectiveness of LLMs for automated chip design, questioning whether their use is merely a gimmick or represents a future trend. Our particular take on this problem is that there is a need to set up open datasets and benchmarks for different problems in chip design, so that the effectiveness and potential use of LLMs can be fully understood and quantified. Our work fits exactly in this category, and we target, in our opinion, the most human labor-intensive part (in terms of engineering) of the chip design process. Our baseline results have demonstrated that LLMs can

achieve satisfactory coverage rates on straightforward designs, but they struggle with more complex ones, suggesting that LLMs do hold promise within the specific context of automated hardware DV.

**Data Asymmetry and LLM4DV as Downstream Evaluation** Owing to the fundamental difference between programming languages used in software and hardware engineering, existing LLMs are presumably more adept with software programming languages like Python and may lack a deep understanding of the semantics of hardware description languages (HDLs). For instance, the StarCoder model's training data comprises various programming languages, yet SystemVerilog and Verilog represent only about $5\%$ of that data (Li et al., 2023). In the meantime, we see the provided LLM4DV flow presents an excellent opportunity to evaluate the capability of LLMs to function as agents for complex tasks, making it an ideal downstream evaluation task.

**Enabling Future DV Research with LLMs** The LLM4DV framework serves as a standard experimentation platform to explore and evaluate DV work. The framework provides an interface for researchers to orchestrate LLMs and input their own prompts for future DV research. For example, advanced approaches, such as coverage-directed generation and mutating tests (Fine and Ziv, 2003b; Guzey and Wang, 2007; Laeufer et al., 2018), could be integrated into the LLM prompts for better coverage. These directions face research challenges that are beyond the scope of this work, but LLM4DV offers an infrastructure on which to build them and baseline results for evaluation.

# 7 CONCLUSION

**@Reviewer 1C9a:** We evaluate LLM4DV using these eight DUT modules and introduce a set of evaluation metrics. Our results show that unoptimized LLMs perform comparably to random guesses in achieving coverage. However, with optimized prompt enhancements, LLMs can achieve coverage rates (a primary metric for measuring verification effectiveness) ranging from 89.74% to 100% in a realistic setting. While these numbers do not surpass those of established formal verification methods, this opens avenues for future research in this direction. We open-source LLM4DV alongside these modules to allow both the machine learning and hardware design communities to experiment with their ideas.

~~We introduce LLM4DV, an open-source benchmark framework designed to efficiently coordinate LLMs for automated hardware test stimuli generation. LLM4DV facilitates integration with diverse DUTs, coverage plans, and LLMs. Our framework has been tested with a range of DUTs and LLMs, and we have developed a set of prompting enhancements that establish solid baselines in the benchmark. Our results illustrate that while these LLMs perform well with simple DUTs, their effectiveness is limited when dealing with more complex designs. This still suggests that LLMs have the potential to overcome common challenges in DV research, such as state space explosion and input specificity, while our framework and benchmarks provide a foundation for exploring and evaluating future DV research. The natural language interface and explainability of LLMs can better integrate domain knowledge into the DV process. We expect that LLM4DV will unlock new research prospects for hardware designers and also serve as a valuable downstream task for assessing LLMs' capabilities for ML researchers.~~

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

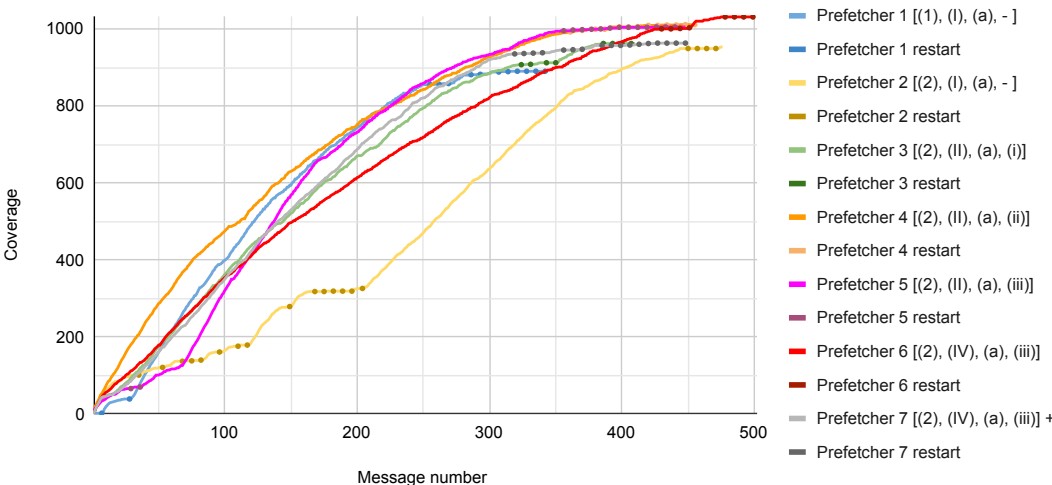

Figure 4: Infinite-message experiments on the Primitive Data Prefetcher Core module. Each line represents the trial reaching the maximum coverage on a configuration, and the dots on it show dialogue restarting points.

# A    APPENDIX

## A.1    DESIGN CHOICES IN PARAMETER SETTINGS

Since the benchmark suite involves chaining multiple rounds of dialogue between the LLM and the testbench framework, we have done a comprehensive evaluation/ablation of the hyperparameters involved to facilitate this agentic behaviour. The majority of these hyperparameters stem from the various prompting engineering techniques/optimizations involved. In fact, many of these prompting techniques are in existing protocols or usage guides on LLMs (Microsoft; OpenAI).

**Maximum message number (700):** We ran a set of "infinite-message experiments" on the Primitive Data Prefetcher Core and Ibex CPU Instruction Decoder, where the maximum response number is much larger than the average exhaustion threshold. As illustrated in the results in Figure 4 and 5, the coverage values of all runs plateau after at most 500 messages, hence we set the maximum message number to 700 for a safety margin.

**Trial termination condition (no hits in 25 messages or fewer than 3 hits in 40 messages):** This prevents over-using the resources when the agent is "exhausted". Extensive tests have demonstrated that LLMs struggle to score additional hits after 25 non-scoring messages or to show significant performance improvement if recording fewer than 3 hits in 40 messages. Typically, we regard this as a "low activity measure" beyond which we ask the model to stop.

**Number of preceding messages (3 responses):** This was determined by considering both the context length restrictions of current LLMs and the typical length of prompts and responses. Maintaining three prompts and responses usually ensures that the maximum context length is not exceeded while retaining as much of the previous dialogue as possible.

**Dialogue restarting tolerance** ($t = 4, 7, 10$)**:** We decide to restart the whole dialogue when the "low activity measure ($< 3$ hits)" is observed in $t$ continuous message queries, since we empirically observe that LLMs' responses can be trapped into local minima. $t$ values are chosen as a comprehensive range in the suitable range below the trial termination condition (since we'd expect to see multiple dialogue restarts before trial termination), with our ablation experiments showing $t = 7$ stably performs the best.

**Number of few-shot examples:** For few-shot prompting, different coverage plans have different "types" of bins. These types are outlined in Appendix A.4. Few-shot prompting is most efficient when one example is given for each bin type. In practice, the number of few-shot samples equals to the bin types for that specific coverage plan.

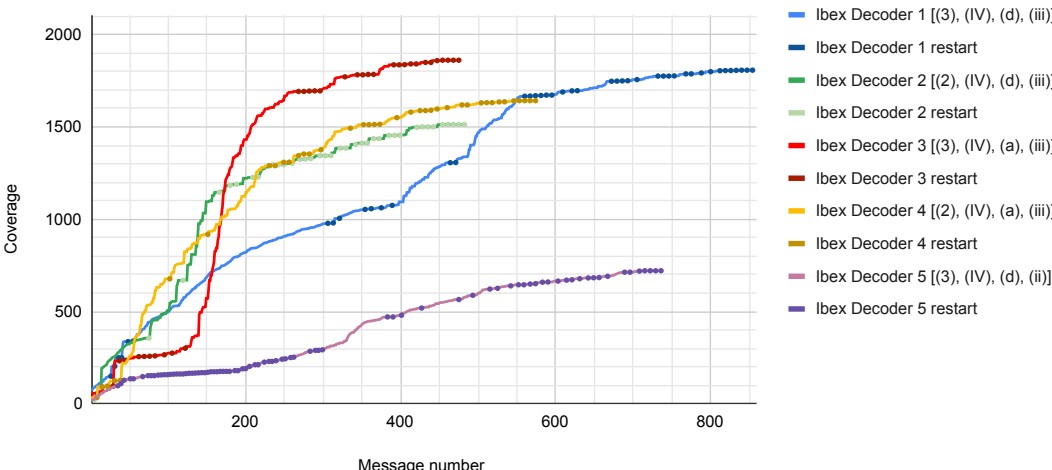

Figure 5: Infinite-message experiments on the Ibex CPU Instruction Decoder module. Each line represents the trial reaching the maximum coverage on a configuration, and the dots on it show dialogue restarting points.

## A.2 Hyperprameter Setup

As outlined in the main text, LLM4DV incorporates a stop condition where if the stimulus generation agent fails to target any bins within 25 responses, or targets fewer than three bins after 40 responses, we consider the agent "exhausted," meaning it is no longer effective or efficient in covering new bins, and the pipeline is halted. These thresholds were determined empirically based on trials with the Primitive Data Perefetcher Core (the simplest Device Under Test) and the Ibex Decoder, alongside experiments involving GPT-3.5.

## A.3 Details of Generic Prompting Improvements

This section describes the design choices of our four prompting improvements.

### A.3.1 Missed-bin sampling

We define three sampling methods:

- (1) **Pure Random Sampling**: randomly samples seven bins from all uncovered bins.
- (2) **Coverpoint Type-based Sampling**: we categorize all bins into "easier bins" and "harder bins" based on their difficulties to be covered, and order them based on their names; when sampling, we always take the first two uncovered bins, then either randomly sample five bins from all uncovered bins if there are no "easier bins" left, or sample three "easier bins" and two "harder bins".
- (3) **Mixed Coverpoint Type-based and Pure Random Sampling**: when the coverage ratio is below 20%, it keeps using Coverpoint Type-based Sampling; when the coverage ratio is larger than 20%, it switches between Coverpoint Type-based Sampling and Pure Random Sampling whenever the current sampling method hits less than three new bins within four responses. The number of 20% is obtained empirically.

### A.3.2 Best-iterative-message sampling

We define four sampling methods:

- (I) **Recent Responses**: keeps the initial query (and its response), and three most recent iterative queries (and their responses).

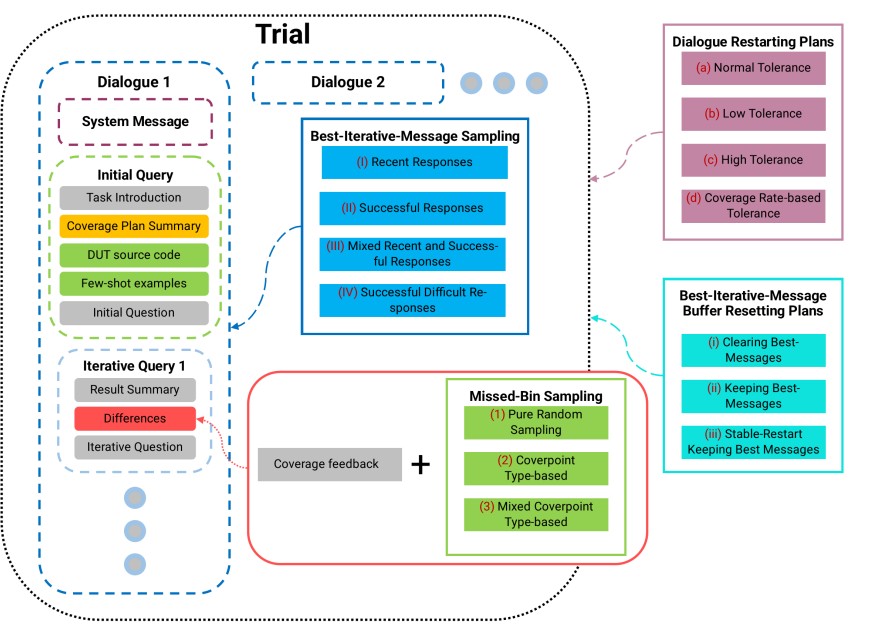

Figure 6: Coverage-feedback prompt template and four prompting improvements. We highlight the different design options possible for the four prompting improvements (DUT source code, Few-shot examples, Best-Iterative-Message Sampling, Dialogue Restarting Plans, Missed-Bin Sampling and Best-Iterative-Message Buffer Resetting Plan).

- **(II) Successful Responses**: keeps the initial query and three iterative queries with responses that hit maximum number of bins; if multiple maximums exist, uniformly samples three of them.
- **(III) Mixed Recent and Successful Responses**: keeps the initial query, two most successful iterative queries, and one most recent iterative query.
- **(IV) Successful Difficult Responses**: similar to successful responses, but each "harder bin" as in Appendix X counts as 2.5 bins.

### A.3.3   DIALOGUE RESTARTING

We define four restarting plans:

- **(a) Normal Tolerance**: restarts the dialogue if the LLM hits less than three bins within 7 responses.
- **(b) Low Tolerance**: restarts the dialogue if the LLM hits less than three bins within 4 responses.
- **(c) High Tolerance**: restarts the dialogue if the LLM hits less than three bins within 10 responses.
- **(d) Coverage Rate-based Tolerance**: uses Low Tolerance when the coverage ratio is below 20%, and Normal Tolerance afterwards. The number of 20% is intentionally set as the same value as in Mixed Coverpoint Type-based and Pure Random Sampling in Appendix A.3.

### A.3.4   BEST-ITERATIVE-MESSAGE BUFFER RESETTING

We define three resetting plans for the best-iterative-message buffer:

- **(i) Clearing best-messages**: clears the buffer on dialogue restarts.
- **(ii) Keeping best-messages**: keeps the buffer on dialogue restarts

- (iii) **Stable-restart Keeping best-messages**: keeps the buffer on dialogue restarts, but not using it for the first four responses after restarts.

## A.4 DETAILS OF DUTS

This section explains the eight DUT modules used and their coverage plans respectively.

### A.4.1 PRIMITIVE DATA PREFETCHER CORE

The Primitive Data Prefetcher Core takes in 32-bit integers and detects whether there's a stride pattern in it. This module requires relatively high mathematical reasoning capability for the LLM.

Our coverage plan contains 1034 bins of the following types:

- Single-stride bins: counts when 16 consecutive integers $a_0, a_1, \ldots, a_{15}$ satisfy $a_{i+1} - a_i = c$ for some constraint $-16 \leq c \leq 15$.
- Double-stride bins: counts when 16 consecutive integers satisfy an alternative stride width pattern, formally $a_{2i+2} - a_{2i+1} = c_1$ and $a_{2i+1} - a_{2i} = c_2$ for some $-16 \leq c_1, c_2 \leq 15$ and $c_1 \neq c_2$.
- Misc bins: including
  - Single-stride positive/negative ocerflow bins: a single stride pattern with $c < -16$ (negative overflow) or $c > 15$ (positive overflow).
  - Double-stride pp/pn/np/nn overflow bins: a double stride pattern with $c_1$ and $c_2$ positively / negatively overflow respectively.
  - No-stride-to-single/double: counts when 16 integers satisfying no stride pattern are followed by 16 integers with single / double stride pattern.
  - Single/double-to-double/single: counts when 16 integers satisfying single / double stride pattern are followed by 16 integers with double / single stride pattern.

### A.4.2 ASYNC FIFO

The Async FIFO is a simple dual clock FIFO, commonly used to transfer data between clock domains. The agent is able to write data using one side, and read data using the other. The simulation is set up so that the write clock has a period of 10ns, while the read clock has a period of 13ns.

Our coverage plan contains 10 bins:

- full_read_wrap: the read pointer wraps to 0.
- gray_read_wrap: the MSB of the read pointer toggles.
- underflow: a read operation is requested while the FIFO is empty
- empty: the FIFO is empty
- read_while_write: on read clock edge a read operation is performed, while a write operation is being requested as well
- full_write_wrap: the write pointer wraps to 0.
- gray_write_wrap: the MSB of the write pointer toggles.
- overflow: a write operation is requested while the FIFO is full
- full: the FIFO is full
- write_while_read: on write clock edge a write operation is performed, while a read operation is being requested as well

### A.4.3 AMPLE PREFETCHER WEIGHT BANK

AMPLE is a GNN FPGA accelerator. The Prefetcher Weight Bank is a small part of the accelerator, responsible for fetching the matrix of weights required to run inference on a fully-connected layer. For the purposes of this investigation, this design can be viewed as a large FIFO. The output of

the device are the contents of the FIFO, which are sent diagonally (i.e. one unit of data is sent from the first row, then from both the first and second, then first, second and third etc.). The two inputs accessed by the LLM agent defines "dimensions" of the output - the maximum number of rows accessed simultaneously, and the number of units of data sent from each row. There is a clear correlation between input values and coverage bins, no significant reasoning is required to achieve full coverage.

Our coverage plan contains 324 bins of the following types:

- in_i: i*16 units of data loaded on each row. Only multiples of 16 can be loaded on each row. If a number that is not divisible by 16 is provided, it will be rounded up by the device.
- out_j: j number of rows loaded with valid data
- combined_features_i_j: i*16 units of data loaded on each row and j number of rows loaded with valid data

### A.4.4 AMPLE PREFETCHER FETCH TAG

AMPLE is a GNN FPGA accelerator. The Prefetcher Fetch Tag is a small part of the accelerator, responsible for fetching the adjacency list, messages and scale factors for a given "node". It includes three queues, an "adjacency queue", a "message queue", and a "scale factor queue". The LLM agent can allocate the Fetch Tag to a node, deallocate it, or load data on one of the queues.

Our coverage plan contains 10 bins:

- adj_dealloc: the DUT is instructed to load the "adjacency queue", but the DUT was not allocated a "nodeslot"
- mess_dealloc: the DUT is insctructed to load the "message queue", but the DUT was not allocated a "nodeslot"
- scale_dealloc: the DUT is insctructed to load the "scale factor queue", but the DUT was not allocated a "nodeslot"
- adj_nomatch: the DUT is insctructed to load the "adjacency queue", but the "nodeslot" provided for this command does not match the "nodeslot" allocated to the DUT
- mess_nomatch: the DUT is insctructed to load the "message queue", but the "nodeslot" provided for this command does not match the "nodeslot" allocated to the DUT
- scale_nomatch: the DUT is insctructed to load the "scale factor queue", but the "nodeslot" provided for this command does not match the "nodeslot" allocated to the DUT
- mess_fetch_adj_nopartial: the DUT is insctructed to load the "message queue", and there is no overflow on the "adjacency queue"
- mess_fetch_adj_partial: the DUT is insctructed to load the "message queue", and there is overflow on the "adjacency queue"
- mess_seen: data is loaded on the "message queue"
- scale_seen: data is loaded on the "scale queue"

### A.4.5 SDRAM CONTROLLER

This SDRAM controller is a simple device that manages the interface to a synchronous dynamic random-access memory (SDRAM), handling tasks such as memory access, data organization, and timing to optimize performance and efficiency.

Our coverage plan contains 7 bins:

- precharge: deactivate (close) the current row of all banks
- auto_refresh: refresh one row of each bank, using an internal counter. All banks must be precharged.
- command_inhibit: command inhibit (no operation)
- load_mode_register: configure the DRAM chip

- activate: open a row for read and write commands

- read: read data from the currently active row

- write: write data to the currently active row

### A.4.6   IBEX INSTRUCTION DECODER

The Ibex Instruction Decoder is an instruction decoder for 32-bit RISC-V instruction codes. This module involves almost no mathematical reasoning but requires knowledge about RISC-V knowledge.

Our coverage plan contains 2107 bins of the following types:

- ALU operation bins: counts when an instruction represents one of 26 pre-defined ALU operations such as ADD, ADDI, XOR, LW, etc.

- Register port bins: counts when an instruction uses the port of the specific register. There are 32 registers, and each has two read ports and one write port, which are used when the register file is taken as the first source, second source, and the destination register, respectively.

- Cross coverage bins: the Cartesian product of ALU operation bins and register port bins. Counts when an instruction satisfies both bins simultaneously (some of the product, such as ADDI and read_port_A of any register, are invalid and not included in the coverage plan).

### A.4.7   IBEX CPU

The Ibex CPU is a full RISC-V CPU. In every cycle the agent provides a stimulus of a list of instructions. Instructions are provided in a sequential manner to the CPU, regardless of the program counter.

Our coverage plan contains 196 bins of the following types:

- Operation bins: for each of pre-defined ten R-type operations, three S-type instruction, and one J-type instruction (JAL), we consider the following four bins:
  - seen: counts when an instruction performs the operation;
  - zero_dst: if available, counts when the instruction performs the operation, with the destination register (rd) as zero (reg #0);
  - zero_src: if available, counts when the instruction performs the operation, with one of the source registers (rs) as zero (reg #0);
  - same_src: if available, counts when the instruction performs the operation, taking the same register as source registers (rs).

- Jump bins: for the JAL operation, we consider forward and backward jumps respectively.

- Hazard bins: for each pair of the pre-defined operations, we consider a simplified read-afterwrite (RaW) hazard, which counts when the later instruction reads from a register that the previous instruction is writing to.

### A.4.8   MIPS CPU

This device is a full MIPS CPU. Similar to the Ibex CPU, every cycle the agent provides a stimulus of a list of instructions. Instructions are provided in a sequential manner to the CPU, regardless of the program counter.

Our coverage plan contains 195 bins of the following types:

- Operation bins: for each of pre-defined ten R-type operations, three I-type instruction, and one J-type instruction (JAL), we consider the following four bins:
  - seen: counts when an instruction performs the operation;
  - zero_dst: if available, counts when the instruction performs the operation, with the destination register (rd) as zero (reg #0);
  - zero_src: if available, counts when the instruction performs the operation, with one of the source registers (rs) as zero (reg #0);

        – same_src: if available, counts when the instruction performs the operation, taking the same register as source registers (rs).

- Jump bins: for the JAL operation, we consider forward and backward jumps respectively.

- Hazard bins: for each pair of the pre-defined operations, we consider a simplified read-afterwrite (RaW) hazard, which counts when the later instruction reads from a register that the previous instruction is writing to.

### A.5 EXAMPLE PROMPTS AND RESPONSES

Figure 7 demonstrates several prompts and responses on the Primitive Data Prefetcher Core module. The agent (USER) introduces the task and coverage plan in the initial message, and then provides coverage feedback in iterative messages. The LLM (ASSISTANT) generates textual responses according to the description and feedback.

### A.6 COMPARISON OF GENERIC PROMPTING IMPROVEMENTS

Due to the cost of money and time for LLM API requests and experiment running, we compare configurations of the stimulus generation agent by their performances using the most promising model (Claude 3 Sonnet) on one of the most complex DUTs (the Ibex CPU). We call the model with parameters as temperature = 0.4, top_p = 1 and max_gen_tokens = 600. These parameters are decided empirically.

All configurations were tested three times. Figure 8 shows the experiment run that achieved maximum coverage for each configuration. The best configuration can be identified as (2) Coverpoint Type-based Sampling, (II) Successful Responses, (a) Normal Tolerance, and (iii) Stable-restart Keeping best-messages, producing a coverage rate of 66.84%.

### A.7 PERFORMANCE WITH AND WITHOUT THE DUT'S SOURCE CODE PROVIDED

Table 6 shows the performance of the LLM models with and without the DUT's source code provided for four designs.

### A.8 RUNTIME COMPARISON OF LLM4DV AND THE FORMAL TOOL

Table 7 shows the runtime comparison of the best trials for each LLM-DUT pair and the formal tool. Both the LLM4DV trials and the formal verification runs were performed on the same machine. Time is reported in seconds.

Table 6: Performance of the LLM models with and without the DUT's source code provided on simpler designs where providing source code is viable. We highlight the **best** results in each case.

| Models | Configurations | Testing Metrics | Asynchronous FIFO | SDRAM Controller | AMPLE Prefetcher Weight Bank | AMPLE Prefetcher Fetch Tag |
|---|---|---|---|---|---|---|
| gpt-3-turbo | Without DUT code | Max coverage | **10 (100%)** | **7 (100%)** | 324 (100%) | **10 (100%)** |
| | | Eff. msg. count | **16** | **7** | 36 | **2** |
| | | Avg. msg. count | **19.7±3.9** | **22.3±11.0** | 50.7±19.3 | **22.0±14.1** |
| | With DUT code | Max coverage | 10 (100%) | 7 (100%) | **324 (100%)** | 9 (90%) |
| | | Eff. msg. count | 19 | 30 | **36** | 32 |
| | | Avg. msg. count | 24.0±7.1 | 30.7±2.5 | **37.7±1.2** | 36.7±3.3 |
| llama-2-70b-chat | Without DUT code | Max coverage | 10 (100%) | 5 (71.43%) | **324 (100%)** | 9 (90%) |
| | | Eff. msg. count | 3 | 28 | **36** | 29 |
| | | Avg. msg. count | 8.3±6.2 | 32.3±4.8 | **41.3±7.5** | 30.3±1.2 |
| | With DUT code | Max coverage | **10 (100%)** | **6 (85.71%)** | 324 (100%) | **10 (100%)** |
| | | Eff. msg. count | **1** | **32** | 48 | **22** |
| | | Avg. msg. count | **10.5±7.9** | **28.3±2.6** | 65.7±18.4 | **27.7±6.0** |
| claude-3-sonnet | Without DUT code | Max coverage | **10 (100%)** | 7 (100%) | **324 (100%)** | **10 (100%)** |
| | | Eff. msg. count | **1** | 6 | **36** | **8** |
| | | Avg. msg. count | **1.0** | 6.7±0.9 | **36.0** | **19.3±8.0** |
| | With DUT code | Max coverage | 10 (100%) | **7 (100%)** | 324 (100%) | 10 (100%) |
| | | Eff. msg. count | 1 | **2** | 36 | 8 |
| | | Avg. msg. count | 1.7±0.9 | **2.3±0.5** | 37.3±1.9 | 19.7±8.3 |
| codellama-70b-instruct | Without DUT code | Max coverage | **10 (100%)** | 7 (100%) | **324 (100%)** | 6 (60.00%) |
| | | Eff. msg. count | **1** | 20 | **44** | 47 |
| | | Avg. msg. count | **3.7±3.1** | 32.0±8.5 | **52.3±8.5** | 40.3±10.9 |
| | With DUT code | Max coverage | 10 (100%) | **7 (100%)** | 324 (100%) | **6 (60%)** |
| | | Eff. msg. count | 1 | **8** | 47 | **34** |
| | | Avg. msg. count | 13.3±9.2 | **29.3±15.1** | 52.7±5.4 | **28.3±4.0** |
| llama-3-70b-instruct | Without DUT code | Max coverage | 10 (100%) | **7 (100%)** | **324 (100%)** | 10 (100%) |
| | | Eff. msg. count | 2 | **1** | **26** | 22 |
| | | Avg. msg. count | 7.7±7.3 | **2.3±1.2** | **32.7±4.7** | 25.0±2.2 |
| | With DUT code | Max coverage | **10 (100%)** | 7 (100%) | 324 (100%) | **10 (100%)** |
| | | Eff. msg. count | **1** | 1 | 28 | **15** |
| | | Avg. msg. count | **1.3±0.5** | 3.0±1.6 | 30.7±3.1 | **20.0±3.6** |

Table 7: Runtime comparison of the best trials for each LLM-DUT pair and the formal tool. Time is shown in seconds, achieved coverage rate is shown in the brackets. Note that the formal tool was given a timeout of 172800s (48 hours).

| | Primitive Data Prefetcher Core | Asynchronous FIFO | AMPLE Prefetcher Weight Bank | AMPLE Prefetcher Fetch Tag |
|---|---|---|---|---|
| gpt-3-turbo | 3312 (98.26%) | 213 (100%) | 1016 (100%) | 30 (100%) |
| llama-2-70b-chat | 10459 (41.68%) | 76 (100%) | 927 (100%) | 156 (100%) |
| claude-3-sonnet | 7753 (77.47%) | 9 (100%) | 761 (100%) | 42 (100%) |
| codellama-70b-instruct | 2456 (7.93%) | 18 (100%) | 854 (100%) | 329 (60%) |
| llama-3-70b-instruct | 12651 (68.67%) | 14 (100%) | 732 (100%) | 142 (100%) |
| Formal verification | 1477 (99.61%) | 51 (100%) | 172800 (0.93%) | 61 (100%) |
| | SDRAM Controller | Ibex CPU Instruction Decoder | Ibex CPU | MIPS CPU |
| gpt-3-turbo | 30 (100%) | 3601 (69.58%) | 503 (19.9%) | 1053 (43.08%) |
| llama-2-70b-chat | 252 (85.71%) | 2447 (19.08%) | 394 (11.22%) | 940 (34.87%) |
| claude-3-sonnet | 19 (100%) | 3359 (71.76%) | 2511 (71.94%) | 4021 (81.54%) |
| codellama-70b-instruct | 77 (100%) | 1981 (19.79%) | 472 (12.76%) | 4758 (46.67%) |
| llama-3-70b-instruct | 24 (100%) | 7881 (53.89%) | 3186 (47.996%) | 8794 (50.26%) |
| Formal verification | 16 (100%) | 1010 (99.95%) | 1087 (100%) | 3181 (100%) |

**SYSTEM :**
Please output a list of (positive or negative) integers only, [...]
Output format: [a, b, c, ...].

**USER 1:**
You will receive a description of bins (i.e. test cases) of a testbench for a hardware [...]
----------
Now, we want to test the DUT with a list of integers as its input [...] Here's the description of the bins that we care about:
------
BINS DESCRIPTION
- The list contains segments (i.e. subsequences) of int.
- Each segment is of length 18.
- A segment follows a single-stride pattern with a stride width x if: the differences between two adjacent integers are always x.
[...]
- For each of the following bins, the list contains at least one segment that satisfies it:
---
  - stride_1_seen - One bin per possible stride width between the minimum and maximum stride width, where the segment follows a single-stride pattern of the stride width.
  - stride_2_seen - One bin per pair of possible stride widths between the minimum and maximum stride width, where the segment follows a double-stride pattern of the stride width pair.
  [...]
---
------
----------
Following the bins description, generate a list that contains segments of integers, which covers the described bins as much as you can.

**ASSISTANT 1:**
To generate a list that covers the described bins, we can start by [...]

Here is a possible solution:

```python
import random

def generate_segment_single_stride(stride_width):
    segment = []
[...]
```

Generated a Python program rather than test stimuli

Generated effective test stimuli

**USER 2:**
The values you provided failed to cover all the bins.
You will see the result coverage of your previous response(s), and then generate another list of integers to cover the unreached bins (i.e. test cases)
Here are some of the unreached bins:
------
UNREACHED BINS
- Single-stride pattern segment of stride width 0 is unreached.
- Single-stride pattern segment of stride width 1 is unreached.
- Single-stride pattern segment of stride width -16 is unreached.
[...]
------
Please regenerate a segment of length 18 for each of these unreached bins according to the BINS DESCRIPTION.

**ASSISTANT 2:**
Apologies for the oversight. Here are the segments to cover the unreached bins:

Segment for single-stride pattern segment of stride width 0:
[0, 0, 0, 0, 0, 0, 0, 0, 0, 0, 0, 0, 0, 0, 0, 0, 0, 0]
Segment for single-stride pattern segment of stride width 1:
[0, 1, 2, 3, 4, 5, 6, 7, 8, 9, 10, 11, 12, 13, 14, 15, 16, 17]
Segment for single-stride pattern segment of stride width -16:
[0, -16, -32, -48, -64, -80, -96, -112, -128, -144, -160, -176, -192, -208, -224, -240, -256, -272]
[...]

Figure 7: Example prompts and responses on the Primitive Data Prefetcher Core module. The purple box is the system message. The green box is an initial query, containing a coverage plan summary (orange). The blue box is an interactive query, containing differences i.e. coverage feedback (red).

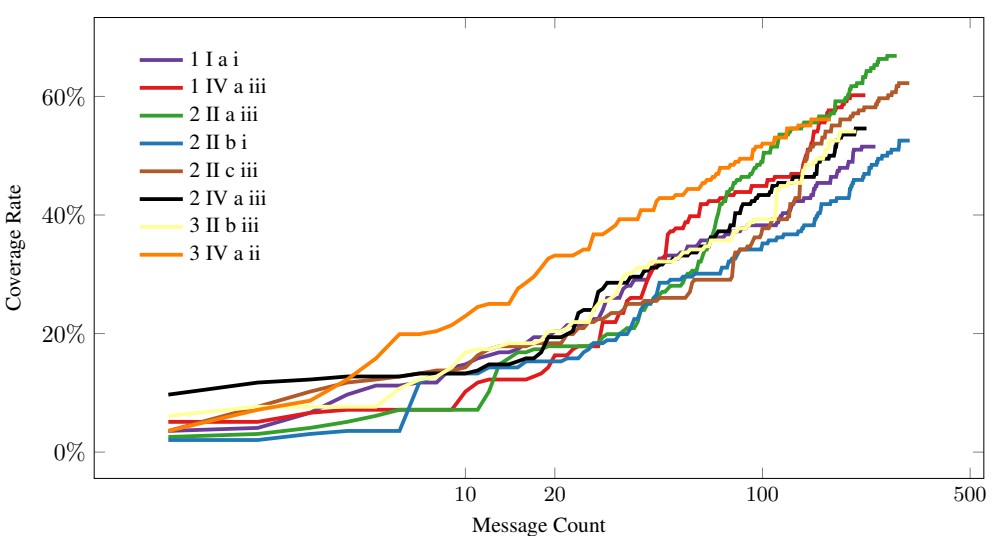

Figure 8: Experiments on the Ibex CPU module. Each line represents the trial reaching the maximum coverage on a configuration. The configurations in legends are illustrated in Figure 6.

