# OpenReview forum: "LLM4DV: Using Large Language Models for Hardware Test Stimuli Generation"
_ICLR.cc/2025/Conference — ICLR 2025 Conference Withdrawn Submission_

### Official Review · Reviewer_RmAo · 2024-11-01

**Soundness:** 2
**Presentation:** 3
**Contribution:** 1
**Rating:** 3
**Confidence:** 4

**Summary:**

This paper proposes a framework using LLM to generate tests for hardware designs. The framework employs a coverage-directed strategy and incorporates well-designed prompting techniques. The authors evaluated the framework on various hardware designs including CPUs. The experimental results show that the framework achieves performance that is comparable to, or better than, traditional approaches such as CRT and formal verification.

**Strengths:**

The authors effectively design the prompting strategies to guide LLMs in generating tests for hardware designs. Additionally, the framework has been tested on various LLM models. The experiments are performed on some industry-level designs such as CPUs, highlighting the impressive capabilities of current LLM technology.

**Weaknesses:**

As a submission to the ICLR conference, the paper lacks novelty in its application of AI concepts, appearing more as an engineering effort. Additionally, there are several similar works on LLMs for hardware test generation that the authors do not mention in this paper, including:

Wang X, Wan G W, Wong S Z, et al. "ChatCPU: An Agile CPU Design & Verification Platform with LLM" (61st Design Automation Conference, 2024).
Qiu R, Zhang G L, Drechsler R, et al. "AutoBench: Automatic Testbench Generation and Evaluation Using LLMs for HDL Design" (2024 International Symposium on Machine Learning for CAD, 2024).
Ma R, Yang Y, Liu Z, et al. "VerilogReader: LLM-Aided Hardware Test Generation" (2024 LLM Aided Design Workshop, 2024).
M. Hassan, S. Ahmadi-Pour, K. Qayyum, C. K. Jha, and R. Drechsler. "LLM-Guided Formal Verification Coupled with Mutation Testing" (2024 Design, Automation & Test in Europe Conference & Exhibition, 2024).

While this submission provides valuable insights into prompting strategies for hardware verification, its contributions appear limited. I recommend that the authors discuss and compare LLM4DV with these related studies and consider submitting the paper to a conference more focused on hardware domains.

**Questions:**

Please compare with recent works in using LLMs for hardware verification.

---

> ### Author Response · Authors · 2024-11-24
>
> We thank the reviewer for the comments and would like to address the following concerns.
>
> > Lacks novelty in application of AI concepts, appearing more as an engineering effort
>
> We would like to point out that many of the papers [2,3,4] mentioned by the reviewer represent concurrent work. We are confident that our contribution is among the earliest in this particular domain.
>
>
> > Related works on LLMs for hardware test generation (4 papers)
> - *ChatCPU* [1]
>     - This work proposes a pipeline for hardware design with iterative HDL generation with LLMs. The pipeline contains a verification flow that uses reference models and unit tests for module-level verification, followed by comprehensive functionality evaluations through co-simulations. However, this paper does not dive into the method of functionality verification and the details of test stimulus generation.
> - *AutoBench* [2]
>     - This work focuses on testbench generation rather than test stimulus generations. This paper is clearly concurrent work.
> - *VerilogReader* [3]
>     - This paper is clearly concurrent work.
> - *LLM-guided Formal Verification Coupled with Mutation Testing* [4]
>     - This work focuses on invariant generation and mutation testing rather than test stimulus generation. Its evaluation is limited to a single DUT with a single LLM (GPT-4). This paper is clearly concurrent work.
>
> 1. Wang X, Wan G W, Wong S Z, et al. "ChatCPU: An Agile CPU Design & Verification Platform with LLM" (61st Design Automation Conference, 2024).
> 2. Qiu R, Zhang G L, Drechsler R, et al. "AutoBench: Automatic Testbench Generation and Evaluation Using LLMs for HDL Design" (2024 International Symposium on Machine Learning for CAD, 2024).
> 3. Ma R, Yang Y, Liu Z, et al. "VerilogReader: LLM-Aided Hardware Test Generation" (2024 LLM Aided Design Workshop, 2024).
> 4. M. Hassan, S. Ahmadi-Pour, K. Qayyum, C. K. Jha, and R. Drechsler. "LLM-Guided Formal Verification Coupled with Mutation Testing" (2024 Design, Automation & Test in Europe Conference & Exhibition, 2024).

---

> > ### Comment · Reviewer_RmAo · 2024-11-28
> >
> > I don't think the authors have addressed my concern and I'll keep my score.

---

### Official Review · Reviewer_1C9a · 2024-11-02

**Soundness:** 3
**Presentation:** 3
**Contribution:** 2
**Rating:** 5
**Confidence:** 3

**Summary:**

The paper proposes LLM4DV, a framework that uses LLMs to generate test stimuli for hardware design verification (DV). The authors then use this framework to evaluate state-of-the-art LLMs on this task. The authors also propose LLM prompting strategies to maximize success in this task. They show that these LLMs achieve test coverage rates ranging from 71.76% to 100%.

**Strengths:**

-	The problem that the authors address is important and challenging.
-	The authors provide an interesting evaluation of SOTA LLMs on the task of hardware design verification.
-	The paper is clear and easy to read.

**Weaknesses:**

While the proposed work is important, the paper has a limited discussion of related work and a limited comparison with existing work (that does not use LLMs). The problem of hardware design verification is well-studied. Examples of papers worth discussing:

-	HIVE: Scalable Hardware-Firmware Co-Verification using Scenario-based Decomposition and Automated Hint Extraction.

-	Survey of Machine Learning for Software-assisted Hardware Design Verification: Past, Present, and Prospect.

As the paper is written now, it is not clear what is its main contribution. Is the main goal of the paper to say that LLMs do better than non-LLM state of the art in the task of DV? Or is the goal to evaluate LLMs on this task (and propose a framework to do so). The authors should clarify this. The paper shows indeed that LLMs can do the task of DV when evaluated using the benchmark proposed by the authors, but it is not written to convince the reader that using LLMs to address this task is better than using other state-of-the-art methods. If your goal is to propose LLMs as the new state-of-the-art in addressing the problem of hardware design verification, you should have a detailed analysis of existing work and should compare with state-of-the-art methods. You also need to use the same benchmarks used by state-of-the-art methods (instead of creating a new benchmark).

The current paper has a high value. It clearly shows that LLMs can perform DV with high coverage rates, but as it is written now it is not convincing enough to show that LLMs are better than existing work in solving the problem.

**Questions:**

Can you clarify what is the goal of the paper? Is the main goal to say that LLMs do better than non-LLM state of the art in the task of DV? Or is the goal to evaluate LLMs on this task (and propose a framework to do so)?

Can you please add a more detailed discussion of existing non-LLM based methods that address the problem of DV? In particular, you can discuss aspects such as their test coverage rates and scalability and compare them to LLM-based methods.

---

> ### Author Response · Authors · 2024-11-24
>
> We thank the reviewer for the comments and would like to address the following concerns. Meanwhile, the revised parts of the paper are colored in blue and the changes reflecting your comments are highlighted with **"@Reviewer 1C9a" in orange**
>
> > Main goal of the paper
>
> We thank the reviewer for mentioning about these related work, our research question is simple and straight-forward in this paper, we are interested in "can LLMs effectively minimize the amount of human effort involved
> in hardware DV?".
>
> To clarify, we explicitly acknowledge that we do not anticipate Large Language Models (LLMs) significantly to outperform existing DV methods. As our results have suggested, LLMs cannot outperform formal verification on certain tasks. Rather, our interest lies in assessing the performance gap between LLMs and standard DV methods like CRT and formal approaches. It is hard for a field (LLM for DV) to make advancements without knowing what is the capability we have at the current state. To this end, our contribution of the paper is to provide a benchmark setup and framework and a set of carefully designed prompting techniques to help establish strong baselines for the DV test stimuli generation task.
>
> We revised the final contribution in our paper to "We evaluate LLM4DV using these eight DUT modules and introduce a set of evaluation metrics. Our results show that unoptimized LLMs perform comparably to random guesses in achieving coverage. However, with optimized prompt enhancements, LLMs can achieve coverage rates (a primary metric for measuring verification effectiveness) ranging from 89.74% to 100% in a realistic setting. While these numbers do not surpass those of established formal verification methods, this opens avenues for future research in this direction. We open-source LLM4DV alongside these modules to allow both the machine learning and hardware design communities to experiment with their ideas."

---

> ### Author Response · Authors · 2024-11-24
>
> > Limited discussion of related work and limited comparison with existing work (that does not use LLMs)
>
> Thanks to the pointer to the related work by the reviewer, we added the comparison with existing non-LLM hardware testing techniques by inserting the following table in the revised version.
>
> | Features       | [1]       | [2]       | [3]       | [4]       | [5]            | [6]            | [7]            | Ours |
> |----------------|----------|----------|----------|----------|----------------|----------------|----------------|------|
> | Models         | Bayesian | Bayesian | Bayesian | Bayesian | DNN            | DNN            | GNN            | LLM  |
> | Encoding model | MAP      | MAP      | MAP      | MAP      | gradient-based | gradient-based | gradient-based | text |
> | Retraining     | Yes      | Yes      | Yes      | Yes      | Yes            | Yes            | Yes            | No   |
>
> We focus on the hardware testing techniques, but also include the results of formal verification as a comparison of interests. All the works in the table exploit ML models to optimize the test genenration process. They encode test generation problem into a representation that can be adopted by their models and require retraining, while our approach benefits the natural text representation to LLMs and do not need to retrain existing models.
>
> We focus on the test generation rather than coverage prediction, so the scalability of all the methods depends on the coverage rates, which affects the simulation count. However, coverage rates are often design-specific, and these works evaluate the effectiveness of their approach on various hardware designs. It is unfair to compare them with our work in coverage rates on different hardware designs. Instead, we provide qualitive comparison with our work in the table above.
>
> The only overlap in DUTs is the IBEX CPU design tested by [7]. They evaluated it on the coverage prediction accuracy and did not provide the coverage rate, while we evaluated it in the covarge rate on test generation.
>
> 1. Shai Fine and Avi Ziv. 2003. Coverage directed test generation for functional verification using Bayesian networks. In Proceedings of the 40th Annual Design Automation Conference. 286–291
> 2. Markus Braun, Shai Fine, and Avi Ziv. 2004. Enhancing the efficiency of bayesian network based coverage directed test generation. In Proceedings of the 9th IEEE International High-Level Design Validation and Test Workshop. IEEE, 75–80.
> 3. Dorit Baras, Shai Fine, Laurent Fournier, Dan Geiger, and Avi Ziv. 2011. Automatic boosting of cross-product coverage using bayesian networks. Int. J. Softw. Tools Technol. Transfer 13, 3 (2011), 247–261.
> 4. Shai Fine, Ari Freund, Itai Jaeger, Yishay Mansour, Yehuda Naveh, and Avi Ziv. 2006. Harnessing machine learning to improve the success rate of stimuli generation. IEEE Trans. Comput. 55, 11 (2006), 1344–1355.
> 5. Raviv Gal, Eldad Haber, Brian Irwin, Marwa Mouallem, Bilal Saleh, and Avi Ziv. 2021. Using deep neural networks and derivative free optimization to accelerate coverage closure. In Proceedings of the ACM/IEEE 3rd Workshop on Machine Learning for CAD (MLCAD’21). IEEE, 1–6
> 6. Raviv Gal, Eldad Haber, and Avi Ziv. 2020. Using DNNs and smart sampling for coverage closure acceleration. In Proceedings of the ACM/IEEE Workshop on Machine Learning for CAD. 15–20.
> 7. Shobha Vasudevan, Wenjie Joe Jiang, David Bieber, Rishabh Singh, C. Richard Ho, Charles Sutton et al. 2021. Learning semantic representations to verify hardware designs. Adv. Neural Info. Process. Syst. 34 (2021).

---

> > ### Author Response · Authors · 2024-11-28
> >
> > Dear Reviewer,
> >
> > We hope that our responses have sufficiently addressed the concerns you raised, especially on addressing our main goals and comparing to related work. We welcome more discussion if you have more questions and suggestions.
> >
> > As the discussion deadline is approaching, we would be very grateful if you could take a moment to review our reply. Thank you for your time and consideration.

---

### Official Review · Reviewer_9kfb · 2024-11-03

**Soundness:** 3
**Presentation:** 3
**Contribution:** 3
**Rating:** 8
**Confidence:** 4

**Summary:**

This work introduces LLM4DV, a framework that uses prompted LLMs to generate test stimuli for hardware Design Verification (DV). The workflow supports a plug-and-play setup, allowing users to experiment with various LLMs, hardware designs, and test coverage plans. The authors develop six prompt enhancements for effective automated DV, establishing strong baselines within LLM4DV as a testbed for studying LLM agentic behavior.

Key contributions include:

A. The creation of three DUT (Design-Under-Test) modules: (1) a Primitive Data Prefetcher Core, (2) an Ibex Instruction Decoder, and (3) an Ibex CPU, and the integration of 5 additional open-source designs, providing a diverse range of DUTs for users.

B. Evaluation of LLM4DV on 8 DUT modules with custom metrics, demonstrating coverage rates from 71.76% to 100% in realistic scenarios with optimized prompts.

Additionally, the authors open-sourced LLM4DV, encouraging both the machine learning and hardware design communities to experiment with it.

**Strengths:**

This research work is highly interesting, offering a promising solution to a critical challenge in hardware design and verification. The manuscript is well-organized and well-written. The LLM4DV framework is thoroughly described, detailing each of its component modules. The evaluation setup is clearly explained, and the results and analysis effectively support the authors' claims. The appendix section provides extensive scientific details that clarify the motivation for benchmarking and the working principles of the framework.

**Weaknesses:**

I have one minor suggestion: Algorithm A.2 should be included in the main text rather than in the appendix, as it would aid in understanding the background more effectively.

**Questions:**

This has already been addressed in the Weaknesses section.

---

> ### Author Response · Authors · 2024-11-24
>
> > Algorithm A.2 should be included in the main text rather than in the appendix, as it would aid in understanding the background more effectively.
>
> We thank the reviewer for the positive comments. We will reorganize the paper to move some important information to the main text in the revised version. The revised parts are colored in blue from line 255 to 287 and the changes reflecting your comment are highlighted with **"@Reviewer 9kfb" in green**.

---

### Official Review · Reviewer_setZ · 2024-11-03

**Soundness:** 3
**Presentation:** 3
**Contribution:** 2
**Rating:** 3
**Confidence:** 4

**Summary:**

The article introduces LLM4DV, an open-source benchmark framework leveraging Large Language Models to automate hardware test stimuli generation, specifically for hardware Design Verification. Hardware DV is crucial for ensuring that hardware designs function correctly, typically involving extensive human input to design and execute test stimuli that cover a range of scenarios. LLM4DV streamlines this process by generating stimuli with LLMs, reducing human effort and improving test coverage.

**Strengths:**

LLM4DV reduces the need for manual testing inputs, saving time and resources. It also Includes multiple evaluation metrics that allow comprehensive assessment beyond just coverage, such as message count efficiency.

**Weaknesses:**

1. I have observed several works with approaches similar to this submission in applying prompting methods to hardware verification and testing. Notably:

*AssertLLM: Generating and Evaluating Hardware Verification Assertions from Design Specifications via Multi-LLMs*

*Towards LLM-Powered Verilog RTL Assistant: Self-Verification and Self-Correction*

*LLM-assisted Generation of Hardware Assertions*

I suggest the author to compare LLM4DV with these works.

2. Novelty Concerns and Venue Suitability:

While the LLM4DV framework in this submission introduces noteworthy adaptations of prompting methods tailored to Hardware Testing, many of these underlying techniques are not novel. Prompting-based methods have been extensively applied in other domains since at least 2022. Specifically:

The Best-iterative-message Sampling described in Section 4.4 resembles a method for summarizing conversational memory, as discussed in Generative Agents: Interactive Simulacra of Human Behavior.

The Few-shot Prompting approach in Section 5 has parallels with work on few-shot learning found in PPT: Pre-trained Prompt Tuning for Few-shot Learning.

3.  Reproducibility and Open-Source Framework Availability:

The authors claim to provide an open-source benchmarking framework, but I was unable to find any code, supplementary materials, or anonymous links for verification. Given my experience with works claiming to be open-source yet only offering minimal or incomplete repositories, I recommend the authors to provide anonymous links during the rebuttal stage would allow me to verify the authenticity and reproducibility of this contribution, which is crucial for enabling follow-up work by future researchers.

**Questions:**

See Weakness.

---

> ### Author Response · Authors · 2024-11-24
>
> We thank the reviewer for the comments and would like to address the following concerns.
> > Compare to several works with approaches similar to this submission in applying prompting methods to hardware verification and testing (3 papers)
>
> The reviewer suggests comparing our approach with three related works that apply prompting methods to hardware verification and testing. Below, we provide detailed distinctions to clarify the unique contributions of LLM4DV.
> - **Comparison with *AssertLLM: Generating and Evaluating Hardware Verification Assertions from Design Specifications via Multi-LLMs* and *LLM-assisted Generation of Hardware Assertions***
>
>   These two works focus on assertion-based verification (ABV), where LLMs are used to generate SystemVerilog Assertions (SVA). While ABV is valuable, it still relies on the availability of test stimuli to activate the assertions and uncover vulnerabilities. In contrast, **LLM4DV targets test stimulus generation**, which is widely recognized as the most labor-intensive stage of hardware verification. This fundamental difference in focus distinguishes our work from the ABV-oriented approaches. We have further discussed the ABV approach in Section 3.
>
> - **Comparison with *Towards LLM-Powered Verilog RTL Assistant: Self-Verification and Self-Correction***
>
>   This work primarily addresses iterative RTL generation rather than test stimulus generation. Although it includes a "self-verification stage" that involves using an LLM to generate a testbench and test cases, it has several notable limitations:
>   (1) The self-verification stage is used as a hinting mechanism for iterative RTL improvement, rather than for conducting comprehensive design verification.
>   (2) There is no guarantee of correctness for the critical LLM-generated testbench, which undermines its reliability.
>   (3) The test cases generated are not systematically constrained or executed in the testbench.
>   As a result, the "self-verification stage" in this work does not fulfill the requirements of authentic design verification. In contrast, LLM4DV is specifically designed to address comprehensive test stimulus generation for verifying designs effectively.
>
> > Many of these underlying techniques are not novel. Prompting-based methods have been extensively applied in other domains since at least 2022
>     - Best-iterative-message Sampling resembles a method for summarizing conversational memory
>     - Few-shot Prompting approach has parallels with work on few-shot learning
>
> The reviewer points out that many of the underlying techniques in our approach are not novel. We acknowledge this, but would like to emphasize how **LLM4DV leverages these methods in a novel and domain-specific way**.
> - **Best-Iterative-Message Sampling v.s. Dialogue Summarization:**
>
>   While Best-Iterative-Message Sampling may resemble methods for summarizing conversational memory, it serves a distinctly different purpose in the context of test stimulus generation. In LLM4DV, the success or failure of generating valid test stimuli often depends on preserving key details from the history messages. In contrast, dialogue summarization typically focuses on condensing conversational context, which can lead to the loss of critical information. Our experiments with dialogue summarization revealed poor preservation of important details, whereas Best-Iterative-Message Sampling buffers recent and successful messages, ensuring that relevant information is retained for accurate stimulus generation.
> - **Few-Shot Prompting in LLM4DV:**
>
>   The few-shot prompting technique indeed has parallels with few-shot learning, but the novelty of our work lies in how we integrate few-shot prompting into a comprehensive, systematic workflow specifically for hardware verification. Our pipeline addresses the labor-intensive challenge of test stimulus generation by incorporating few-shot prompting, alongside other complementary techniques, and conducting extensive experiments to evaluate their effectiveness on this domain- specific task. Furthermore, we have designed workarounds to handle emerging challenges, making our approach both practical and effective for the hardware verification domain.
>
> One core message we would like to convey is the significant influence of prompting techniques on the performance of LLMs in achieving coverage rates. Without such treatments, LLMs seldomly exceed the effectiveness of random guesses. Through extensive testing, we have applied a wide array of prompting methods, many of which are already established. Yet, the contribution of our study is in showing that **optimized prompting can elevate the performance of existing LLMs to levels comparable with strong baselines, such as CRT.** We believe this finding is both valuable and novel for the hardware DV community.

---

> ### Author Response · Authors · 2024-11-24
>
> > I recommend the authors to provide anonymous links during the rebuttal stage would allow me to verify the authenticity and reproducibility of this contribution
> - Here is a link to our anonymized repository: https://github.com/d47e752/llm4dv

---

> > ### Comment · Reviewer_setZ · 2024-12-02
> > **Thanks for the reply!**
> >
> > I appreciate the author's response. In their reply, the author also acknowledged the lack of innovation in terms of algorithms. Therefore, I still believe that this paper should be submitted to a hardware-focused conference. ICLR may not be the best venue for this work. I will maintain my score.

---

> > > ### Author Response · Authors · 2024-12-02
> > >
> > > We appreciate the reviewer's feedback. We would like to emphasize that although prompting techniques are pre-existing, we apply them innovatively within a domain-specific context. More importantly, no open benchmarks or datasets presently exist for this line of research, making ours the work offering an open evaluation of this particular downstream problem.

---

### Official Review · Reviewer_M6Uz · 2024-11-06

**Soundness:** 2
**Presentation:** 2
**Contribution:** 2
**Rating:** 3
**Confidence:** 4

**Summary:**

The paper presents LLM4DV, an open-source benchmarking framework designed to automate hardware test stimuli generation using large language models (LLMs) for hardware design verification (DV). LLM4DV leverages six LLMs and introduces six prompt engineering improvements to tackle the labor-intensive DV process, aiming to enhance coverage by generating stimuli for various hardware devices under test (DUTs). The framework evaluates these LLMs across eight hardware designs, comparing their coverage rates and effectiveness against conventional constrained-random testing (CRT) and formal verification methods. While LLM4DV demonstrates notable performance improvements on simpler DUTs, its efficacy declines with complex hardware architectures due to limitations like state-space explosion, token window constraints, and a high dependency on prompt engineering. The study concludes that LLMs show promise for automating aspects of DV but require further adaptation to handle more complex verification scenarios effectively.

**Strengths:**

The paper tries to address a relevant problem and is well written, and well organized.

**Weaknesses:**

1) The LLM4DV framework relies on missed-bin sampling to manage the large number of uncovered coverage bins, but this approach introduces randomness that may hinder consistent exploration of harder-to-hit bins. This sampling method may lead to poor prioritization, with the LLM potentially focusing on simpler bins at the expense of complex corner cases that are critical for comprehensive hardware verification.
2)	The dialog restarting mechanism, designed to mitigate the issue of LLMs "getting stuck" by clearing dialog history, lacks adaptability. The fixed thresholds (like hitting fewer than three bins within 40 messages) are rigid and may not accommodate variability across hardware designs with different complexity levels. This static approach may cause premature restarts, losing valuable accumulated context and decreasing efficiency in achieving coverage.
3)	Including DUT source code in the prompt often exceeds the token limit for many LLMs, especially in the case of designs with complex logic or multiple functional modules. This limitation results in truncating prompts or excluding critical code portions, reducing the model’s ability to effectively map test stimuli to specific coverage bins.
4)	The framework’s effectiveness heavily depends on manually engineered prompts and specific configurations, such as coverage-feedback templates and best-iterative-message sampling. This approach introduces overhead and suggests a lack of automated prompt optimization, which could otherwise make LLM4DV more adaptable and less reliant on human intervention.
5)	For designs with large state spaces, such as components with storage elements or multiple queues (e.g., AMPLE Prefetcher Weight Bank), the LLM4DV framework struggles to maintain efficient coverage. This is because the LLM’s state exploration strategy does not scale well with increasing design complexity, leading to a "state-space explosion" where the LLM cannot feasibly cover all unique states or transitions.

**Questions:**

1) Given the fixed thresholds in dialog restarting, how does LLM4DV adapt to variability in coverage rates across designs with differing complexity? Could dynamic thresholds improve restart efficiency?
2)	For designs with detailed CPU architectures, like those with complex instruction sets (e.g., RISC-V), how does LLM4DV ensure that test stimuli generation adequately captures inter-instruction dependencies to achieve comprehensive coverage?
3)	Does LLM4DV include mechanisms to help identify or diagnose the root causes of missed coverage bins? How could such a feature enhance the utility of the framework for iterative debugging workflows?

---

> ### Author Response · Authors · 2024-11-24
>
> We thank the reviewer for the comments and would like to address the following concerns.
>
> > Missed-bin sampling introduces randomness that may hinder consistent exploration of harder-to-hit bins. May lead to poor prioritization, with the LLM potentially focusing on simpler bins at the expense of complex corner cases that are critical for comprehensive hardware verification
>
> The reviewer raises a valid concern about the potential impact of missed-bin sampling randomness on the consistent exploration of harder-to-hit bins and the risk of deprioritizing critical corner cases.
>
> Our experiments, however, indicate that **consistent attempts at harder-to-hit bins do not lead to improved exploration by LLMs**. Instead, LLMs tend to repeatedly make the same errors when targeting these challenging bins. To mitigate this, we **intentionally introduce randomness** in the sampling process to encourage exploration of other bins. This strategy enables the LLM to gain valuable insights and experience from successful hits, which can later be leveraged to revisit and overcome the repeated mistakes in harder-to-hit bins.
>
> This balance between exploration and exploitation ensures that LLM4DV does not stagnate on a subset of bins while maintaining the potential to address critical corner cases effectively.
>
> > Dialog restarting mechanism lacks adaptability. Fixed thresholds may not accommodate variability across hardware designs with different complexity levels. May cause premature restarts, losing valuable accumulated context and decreasing efficiency. Given the fixed thresholds in dialog restarting, how does LLM4DV adapt to variability in coverage rates across designs with differing complexity? Could dynamic thresholds improve restart efficiency?
>
> - **Preserving Valuable Context During Dialogue Restarts:**
>   To retain valuable accumulated context while preventing LLMs from repeating prior mistakes, we employ the **Best-iterative-message Buffer Reset strategies**. This approach uses a buffer that stores three messages that achieved most hits (see Best-iterative-message Sampling in Section 4.4). By using "Keeping Best-Messages" or "Stable-Restart Keeping Best-Messages" during restarts, the buffer is preserved across dialogue resets, enabling the accumulation of meaningful context while avoiding redundant errors.
> - **Evaluation of Adaptive Restarting Thresholds:**
>   We conducted experiments with adaptive restarting thresholds based on current coverage and recent hits. As shown in Figure 5 (Appendix), the Ibex Decoder 1 coverage curve applies an adaptive restarting threshold, while the Ibex Decoder 3 curve uses a normal restarting threshold. The results indicate that the adaptive restarting threshold underperformed compared to the normal threshold in both Max Coverage and Effective Message Count metrics.
> - **Challenges of Automating Adaptive Restarting Thresholds Across Trials:**
>   Automating adaptive dialogue restarting thresholds across trials (as opposed to across messages) is challenging due to the interplay of multiple factors. For example, poor coverage could result from repeated mistakes on harder bins, in which more frequent restarts might mitigate wasted attempts on these errors, but hinder the LLM from learning from the mistakes. Without manual interpretation to identify the root cause of poor coverage, the system would rely on trial-and-error to update the threshold, which is computationally expensive and often impractical.

---

> ### Author Response · Authors · 2024-11-24
>
> > Including DUT source code in the prompt often exceeds the token limit for many LLMs, resulting in truncating prompts or excluding critical code portions
> >
> > The framework’s effectiveness heavily depends on manually engineered prompts and specific configurations, such as coverage-feedback templates and best-iterative-message sampling. This approach introduces overhead and suggests a lack of automated prompt optimization, which could otherwise make LLM4DV more adaptable and less reliant on human intervention.
>
> The reviewer highlights two key concerns: the challenges posed by LLM token limits when incorporating DUT source code and the reliance on manually engineered prompts and configurations.
> - **Addressing Token Limitations:**
>   The limited context length of LLMs is indeed a significant constraint that influences our approach to incorporating knowledge of the DUT. As detailed in Section 4.5, directly providing the DUT source code is only feasible when its length is within the token limits. For larger DUTs, we opt to manually provide a concise task description, as outlined in Section 4.3. This approach ensures a reliable and efficient summary of the DUT code and coverage bins, allowing us to retain the most critical information while working within the LLM’s context constraints.
> - **Managing Configuration Overhead:**
>   While it is true that LLM4DV relies on specific configurations and engineered prompts, these are the result of extensive experimentation and optimization (Appendix A.6). As described in Section 5, the best-performing configuration comprises **Coverpoint Type-based Sampling, Successful Responses, Normal Tolerance, and Stable-restart Keeping best-messages**. The results in Section 5 demonstrate the robustness and versatility of this setup, showcasing broad applicability across diverse LLMs and DUTs.
>   Moreover, the adaptability of LLM4DV allows future integration of automated prompt optimization techniques as LLM capabilities and tooling evolve, further reducing reliance on manual interventions.
>
> > For designs with large state spaces, such as components with storage elements or multiple queues (e.g., AMPLE Prefetcher Weight Bank), the LLM4DV framework struggles to maintain efficient coverage. This is because the LLM’s state exploration strategy does not scale well with increasing design complexity, leading to a "state-space explosion" where the LLM cannot feasibly cover all unique states or transitions.
> - The reviewer raises a valid concern regarding the challenges of state-space explosion. However, **LLM4DV demonstrates superior capability in addressing the state-space explosion problem** compared to formal verification methods. For instance, as shown in Table 4, LLM4DV achieved **100%** coverage of the AMPLE Prefetcher Weight Bank across all tested LLMs, whereas the formal verification method reached only **0.93%** coverage. This result highlights the scalability and efficiency of LLM4DV in exploring complex state spaces effectively.
>
> > For designs with detailed CPU architectures, like those with complex instruction sets (e.g., RISC-V), how does LLM4DV ensure that test stimuli generation adequately captures inter-instruction dependencies to achieve comprehensive coverage?
>
> The reviewer raises an important question about how inter-instruction dependencies are captured. LLM4DV addresses this challenge in multiple ways:
> - **Leveraging Pretrained Knowledge:** The LLM utilizes its pretrained knowledge of the ISA (e.g., RISC-V) and common inter-instruction dependencies (e.g., data hazards in pipelined CPUs) to generate test stimuli and refine them iteratively for hard-to-hit bins.
> - **Plug-and-Play Flexibility:** A key goal of introducing LLMs to the DV pipeline is to harness their pretrained knowledge and reasoning capabilities to generate test stimuli at scale, reducing the reliance on manual effort. Our pipeline is designed to be plug-and-play to accommodate future improvements in model pretraining and capabilities, enabling seamless integration of different LLMs.
> - **Enhanced Problem-Solving via Iterative Feedback:** We enhance the LLM's ability to tackle complex DV challenges through a combination of DUT-specific hints, coverage bin descriptions, few-shot prompting, and iterative feedback from the testbench. These strategies help the LLM refine its approach and better address intricate dependencies.
>
> These measures collectively ensure that LLM4DV achieves comprehensive coverage for complex CPU architectures, addressing the inter-instruction dependency concerns effectively.

---

> ### Author Response · Authors · 2024-11-24
>
> > Does LLM4DV include mechanisms to help identify or diagnose the root causes of missed coverage bins? How could such a feature enhance the utility of the framework for iterative debugging workflows?
>
> We would like to clarify for the reviewer that our objective is not to solve the problem of generating DV test stimuli, as there are inherent limitations to the current capabilities of LLMs. Instead, our goal is to curate a set of tasks within this domain and to create a robust baseline for these proposed tasks.

---

> > ### Author Response · Authors · 2024-11-28
> >
> > Dear Reviewer,
> >
> > We hope that our responses have sufficiently addressed the concerns you raised. We welcome more discussion if you have more questions and suggestions.
> >
> > As the discussion deadline is approaching, we would be very grateful if you could take a moment to review our reply. Thank you for your time and consideration.

---

### Note · Authors · 2025-01-10

I have read and agree with the venue's withdrawal policy on behalf of myself and my co-authors.